# Membrane characteristics tune activities of endosomal and autophagic human VPS34 complexes

Yohei Ohashi[1†], Shirley Tremel[1†], Glenn Robert Masson[1], Lauren McGinney[1], Jerome Boulanger[1], Ksenia Rostislavleva[1], Christopher M Johnson[1], Izabella Niewczas[2], Jonathan Clark[2], Roger L Williams[1]*

[1]MRC Laboratory of Molecular Biology, Francis Crick Avenue, Cambridge, United Kingdom; [2]Babraham Institute, Cambridge, United Kingdom

**Abstract** The lipid kinase VPS34 orchestrates diverse processes, including autophagy, endocytic sorting, phagocytosis, anabolic responses and cell division. VPS34 forms various complexes that help adapt it to specific pathways, with complexes I and II being the most prominent ones. We found that physicochemical properties of membranes strongly modulate VPS34 activity. Greater unsaturation of both substrate and non-substrate lipids, negative charge and curvature activate VPS34 complexes, adapting them to their cellular compartments. Hydrogen/deuterium exchange mass spectrometry (HDX-MS) of complexes I and II on membranes elucidated structural determinants that enable them to bind membranes. Among these are the Barkor/ATG14L autophagosome targeting sequence (BATS), which makes autophagy-specific complex I more active than the endocytic complex II, and the Beclin1 BARA domain. Interestingly, even though Beclin1 BARA is common to both complexes, its membrane-interacting loops are critical for complex II, but have only a minor role for complex I.

*For correspondence:
rlw@mrc-lmb.cam.ac.uk

†These authors contributed equally to this work

Competing interests: The authors declare that no competing interests exist.

## Introduction

Lipids form membranes that delimit cellular compartments, but they also can act as signalling molecules that recruit proteins to membranes. The chemical and compositional diversity of lipids help to establish the identity of intracellular compartments and the nature of subdomains within these compartments. For example, the endoplasmic reticulum (ER) contains low levels of cholesterol and high levels of unsaturated glycerophospholipids, whereas the plasma membrane has an abundance of cholesterol and saturated sphingolipids and glycerophospholipids (*Bigay and Antonny, 2012*). Differences in lipid composition give membranes their physicochemical characteristics, which can control the activities of membrane-associated enzymes. The lipid phosphatidylinositol 3-phosphate (PI(3)P) plays important roles in the regulation of various intracellular events including autophagy, endocytosis, phagocytosis, and vacuolar protein sorting. In mammals, PI(3)P is generated mainly by the Class III phosphatidylinositol 3-kinase VPS34/PIK3C3, which phosphorylates the 3-OH position of phosphatidylinositol (PI). However, in cells, VPS34 is not found alone. Instead, it forms two principal complexes, complexes I and II. Both complexes have core components VPS34, VPS15 and Beclin 1 (Vps30 in yeast) bound to a specific component, which is either ATG14L (Atg14 in yeast) in complex I or UVRAG (Vps38 in yeast) in complex II. This single subunit difference between these two complexes localizes them to different membrane compartments. Complex I is active on autophagosomes (*Matsunaga et al., 2009*; *Sun et al., 2008*; *Itakura et al., 2008*) and some ER compartments (*Hamasaki et al., 2013*; *Nascimbeni et al., 2017*; *Axe et al., 2008*), in response to starvation. In contrast, complex II is mainly active in endocytic pathways by partially co-localizing in mammalian cells with Rab5- and Rab7-positive endolysosomal compartments as well as with Rab9-positive

compartments important for endosome to TGN sorting (*Itakura et al., 2008*). As the membranes of these different compartments vary, we have undertaken to determine how lipid composition affects the lipid kinase activity of VPS34 complexes.

The PI(3)P generated by complex I appears in puncta at ER-mitochondria contact sites, at ER-plasma membrane contact sites and on ER sites known as omegasomes, which are recognized by PI(3)P-specific binding proteins such as DFCP (*Nascimbeni et al., 2017*; *Prinz et al., 2020*; *Hamasaki et al., 2013*; *Axe et al., 2008*). These findings indicate that complex I is recruited to specific sites on membranes for its activation. This recruitment involves an upstream complex, the ULK1 complex (Atg1 complex in yeast) (*Russell et al., 2013*). However, how complex I itself associates with membranes and is activated on them is unclear. Complex II recruitment to early endosomes and activation is aided by Rab5 (*Christoforidis et al., 1999*; *Murray et al., 2002*), although the mechanism by which this occurs is poorly characterized. PI(3)P produced by complex II, in turn, leads to recruitment of various PI(3)P binding proteins that function in endocytic pathways such as Hrs, EEA1, and sorting nexins (*Gaullier et al., 1999*; *Xu et al., 2001*). We have previously shown that yeast complex II has a Y-shaped structure, with the one arm bearing Vps34 and Vps15 acting as the catalytic arm, while the other arm bearing Vps30 and Vps38 is the adaptor arm (*Rostislavleva et al., 2015*). A similar structural organization has been observed for human complexes I and II (*Young et al., 2019*; *Baskaran et al., 2014*; *Ma et al., 2017*). The aromatic finger motif in the BARA domain of Vps30 is important for membrane binding and kinase activity of complex II, shedding light on the importance of the adaptor arm for membrane binding and kinase activity. Many previous *in vitro* studies of Class III PI3K have used pure PI (*Fassy et al., 2017*; *Lu et al., 2014*; *Munson and Ganley, 2016*; *Russell et al., 2013*; *Zhong et al., 2009*; *Kim et al., 2013*) or a lipid mixture of only PI and PS as the substrate (*Brier et al., 2019*). This approach underestimates the effects that membrane complexity could have on VPS34 activity. For this reason, we have devised alternative VPS34 assays to measure the activities of Class III PI3Ks on membranes whose character we can control.

Protein binding to membranes is affected by three important physicochemical parameters: membrane electrostatics, lipid packing, and membrane curvature (*Bigay and Antonny, 2012*). The negative charge of intracellular membranes arises mainly from phosphoinositides and phosphatidylserine (PS). PS is known to enhance membrane-protein interactions by binding to basic residues of proteins (*Xu et al., 2013*; *Yeung et al., 2009*; *Yeung et al., 2006*; *Zhu et al., 1999*), whereas protein affinity for a phosphoinositide may depend on both the net charge (the number of phosphates on the inositol ring) or stereochemistry of the phosphoinositide.

Lipid packing is determined by the shape of polar lipid headgroups and by saturation of acyl chains. Unsaturated acyl chains only loosely pack into membranes due to the kinks introduced by unsaturated bond(s), whereas completely saturated acyl chains provide a more uniform geometry and can therefore pack tightly to restrict membrane flexibility (*Bigay and Antonny, 2012*). Packing also depends on global membrane properties, such as stretching/compression, membrane curvature, or membrane stress (*Johner et al., 2014*; *Marsh, 2007*; *McMahon and Boucrot, 2015*; *Bigay and Antonny, 2012*). Membrane flexibility is important for processes such as vesiculation, endocytosis and phagocytosis (*Calder et al., 1990*; *Degreif et al., 2019*; *Jackson, 2018*; *Schroit and Gallily, 1979*; *Manni et al., 2018*). Phospholipids usually have a saturated fatty acyl chain at position *sn1* of the glycerophospholipid, while the *sn2* fatty acyl chain can be saturated, monounsaturated or polyunsaturated. Phospholipids with two polyunsaturated fatty acyl chains make membranes more flexible, but very permeable, while the more common *sn1*-saturated-*sn2*-polyunsaturated enable flexibility with low membrane permeability (*Manni et al., 2018*). Acyl chain saturation status also influences membrane curvature. Unsaturated lipids can tolerate high membrane curvature, whereas the shapes of saturated lipids are cylindrical, making membranes containing them flatter. ER membranes are enriched with unsaturated phospholipids, enabling high membrane curvature. This facilitates protein secretion from the ER. In contrast, PM membranes are more tightly packed, due to the abundance of saturated phospholipid, which creates a thick boundary between the cell and its environment (*Holthuis and Menon, 2014*; *Bigay and Antonny, 2012*).

Here, we have developed a purely *in vitro* system with purified VPS34 complexes I and II in which we can directly control membrane bilayer properties, including acyl chain saturation of both substrate and non-substrate lipids, membrane charge and membrane curvature. By combining HDX-MS with mutagenesis and lipid kinase assays, we have investigated which domains and motifs of

complex I and complex II are important for membrane binding and kinase activity, and which physicochemical parameters of membranes have the greatest impact on complexes I and II.

## Results

To assess how membrane properties modify the activities of VPS34 complexes, we have developed a quantitative confocal microscopy assay to measure the activity of the human Class III PI3K complexes on giant unilamellar vesicles (GUVs), which have a radius of ~ 1–30 μm. Purified, recombinant, monodisperse VPS34 complexes I or II (*Figure 1A*) were incubated with Lissamine-Rhodamine labelled GUVs containing PI and an AlexaFluor647-labelled PX domain, which specifically binds PI(3)P (*Ellson et al., 2001*; *Figure 1B*). By following the rate of recruitment of the PX domain to the GUVs, we can estimate the PI(3)P production and compare the relative activities of the complexes on a range of membrane types.

### Human complex I is more active than complex II

We first compared the activities of complexes I and II on membranes containing 18% PI (with a mixed chain distribution, purified from bovine liver, hereafter called mixed chain PI), 10% DOPS, 17% DOPE, and 55% DOPC (see *Supplementary files 1* and *2*). This lipid headgroup composition resembles the ER membrane and we will refer to it as the base lipid composition or 'DO base'. On DO base lipids, human complex I showed higher activity than complex II. Complex I showed a 7-fold higher initial rate compared with complex II (*Figure 1C*). Next, we examined whether the difference in activity between complexes I and II could be because complex II, which is normally active on early endosomes, was not assayed on endosome-mimicking membranes. There is evidence that early endosomes are rich in cholesterol and sphingomyelin (*Bissig and Gruenberg, 2013*; *Arumugam and Kaur, 2017*; *Kobayashi et al., 2002*). Therefore, we measured the activity of complexes I and II in the presence of 10% cholesterol and 10% sphingomyelin (*Figure 1—figure supplement 1*). This showed that even with this membrane composition, complex I remained more active than complex II. In order to compare the ability of complexes I and II to bind membranes, we carried out a flotation assay (*Figure 1D*). This revealed that fractions 1–3, which contained floating liposomes, included much more complex I when compared to the same fractions with complex II (*Figure 1D*). The correlation of the activity assays with the membrane binding assay strongly suggests that complex I is more active than complex II because complex I binds membranes more tightly than complex II.

### Both substrate and non-substrate lipid unsaturation as well as membrane curvature greatly increase VPS34 activities

We examined how lipid packing affects VPS34 activities by varying the saturation status of the acyl chains. For each complex, we compared VPS34 activity on GUVs composed of different percentages of phospholipids with either stearoyl (18:0) or oleoyl (18:1) acyl chains (*Figure 2A*). As stearoyl acyl chains are saturated, they cause a tighter lipid packing than the mono-unsaturated oleoyl chains. Whereas SO lipids are made with one stearoyl and one oleoyl acyl chain, DO lipids carry two oleoyl acyl chains. Three different lipid compositions were examined: 82% DO lipids (18:1-18:1), 82% SO lipids (18:0-18:1), and 55% SO with 27% DO lipids (55% SO+27% DO) (*Figure 2B*). In each case, we varied only the acyl chain saturation of the non-substrate lipids while we kept the source of the PI and the relative proportions of PI, PS, PE and PC the same. GUV assays revealed that complexes I and II were only active on 82% DO lipids and not on 55% SO+27% DO or 82% SO (*Figure 2C and D*), suggesting that loose lipid packing activates complexes I and II. However, Z-stack analysis of ten Z slices, which covers a range of vesicle sizes revealed some activity on smaller vesicles with 55% SO +27% DO, but not on vesicles with 82% SO (*Figure 2E and F*). To examine this further, the curvatures (1/radius) for each GUV mixture were plotted against their fluorescence intensities. GUVs with 55% SO+27% DO displayed a positive correlation between increasing membrane curvature and activity for both complexes I and II, but 82% SO GUVs did not exhibit this correlation (*Figure 2G and H*). This suggests that high membrane curvature can at least partially compensate for the negative effect of lipid saturation.

We also examined the effect of lipid saturation of the substrate PI, while not changing the surrounding non-substrate lipids made of DO (*Figure 3A*). Four PI species were examined: DSPI (18:0-

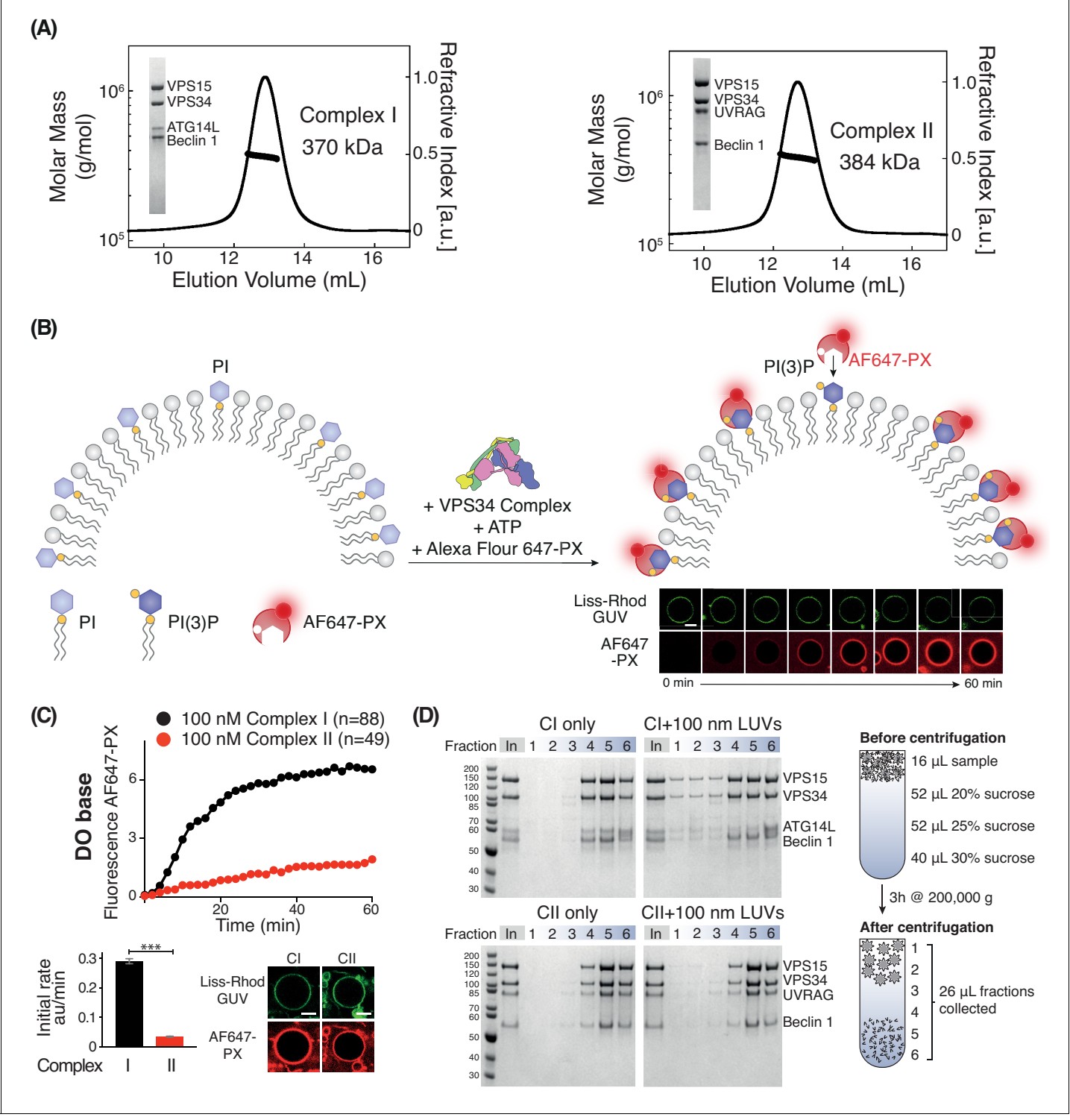

**Figure 1.** Activities of human complexes I and II on GUVs. (**A**) SEC-MALS analysis of purified human Class III PI3K complex I (left panel) and complex II (right panel). The protein samples were run on a Superose 6 10/300 column. Insets: SDS-PAGE of starting material stained with Coomassie staining. Both complex I and complex II are monodisperse, with an average mass for complex I consistent with a 1:1:1:1 Beclin 1/VPS15/VPS34/ATG14L complex (370 kDa, theoretical mass: 363 kDa) and an average mass for complex II consistent with a 1:1:1:1 Beclin 1/VPS15/VPS34/UVRAG complex (384 kDa, theoretical mass: 386 kDa). (**B**) Assay design for activities of human complexes I and II on GUVs, using confocal microscopy and a labelled PI(3)P-binding p40-PX domain (AF647-PX). Fluorescence from the Lissamine-Rhodamine (Liss-Rhod GUV) channel delineates the membrane, while the AF647 is indicative of VPS34 activity on the membrane. Scale bar: 5 μm. (**C**) Complex I is more active than complex II on GUVs with 'DO base' lipids (18%

*Figure 1 continued on next page*

*Figure 1 continued*

mixed chain PI, 10% DOPS, 17% DOPE and 55% DOPC). In the lower panels, the initial rates after a lag phase (AF647-PX fluorescence change/min in arbitrary units, AU) and confocal images corresponding to the AF647-PX and Liss-Rhod channels at the end of the reaction are shown. For clarity, only mean values of measurements for each time point in the reaction progress curves are plotted here and throughout all figures. Plots with SDs for each time point are shown in **Supplementary file 3**. All scale bars: 5 µm. ***: p<0.001 (p<0.0001). **Figure 1—figure supplement 2** illustrates some of the raw images that were used for quantification. Scale bars: 15 µm. (D) Membrane binding of complexes I and II using a lipid flotation assay. Large unilamellar vesicles (LUVs, 100 nm) and proteins were mixed and pipetted on a sucrose gradient. The gradient was then centrifuged, fractionated and analysed by SDS-PAGE. Fractions 1–3 are the least dense fractions of a sucrose gradient, containing floating vesicles and membrane-bound proteins. Fractions 5–6 are the highest density sucrose fractions, containing the pelleted proteins. The complexes alone sediment into the denser portion of the gradients (left), but the presence of LUVs cause the complexes to associate with LUVs floating on the top of the gradient (right). Gel quantification can be seen in **Figure 1—figure supplement 3**.

The online version of this article includes the following figure supplement(s) for figure 1:

**Figure supplement 1.** Activities of complexes I and II on GUVs containing cholesterol and sphingomyelin.
**Figure supplement 2.** Selected images of GUVs for the time courses shown in *Figure 1C*.
**Figure supplement 3.** Quantification of membrane binding of complex I (**A**) and complex II (**B**) using a lipid flotation assay (*Figure 1 (D)*).

18:0), SAPI (18:0-20:4), DOPI (18:1-18:1) and the mixed chain PI (*Figure 3A*). GUV assays with complex I showed that DOPI was the best substrate, followed by the mixed chain PI and SAPI. However, with the more saturated acyl chains of DSPI, the activity of complex I was greatly reduced (*Figure 3B*).

As a control, we tested whether lipid packing could affect binding of the AF647-PX reporter domain to PI(3)P. We found that GUVs with 3% DOPI(3)P in 82% SO lipids showed a similar PX binding as the GUVs with 3% DOPI(3)P in 82% DO lipids (*Figure 3C*), indicating that membrane packing had little effect on PX binding. The binding of the PX reporter to DSPI(3)P and SOPI(3)P containing GUVs was not tested, since there is no commercial source of these lipids.

Together, these results show that lipid packing affects the complexes I and II activities, with both complexes showing a dramatic preference for unsaturated lipids, in both the general background lipids of the membrane and in the PI substrate for the enzyme. They also show that the negative impact of tight lipid packing on activities of VPS34 complexes can be partially compensated by increased membrane curvature.

## Increasing negative membrane charge activates complex I, complex II, and VPS34 alone

In order to examine the effect of negative membrane charge on VPS34 activity, we made membranes with increasing proportions of the negatively charged lipid PS. We compared activities on GUVs with 10% PS (DO base lipids) and 25% PS (DO high PS lipids) (*Figure 4A and B*). Using VPS34 alone, we could see no activity on GUVs with 10% PS. However, on GUVs with 25% PS, VPS34 activity was clearly detectable (*Figure 4C*). Also, complexes I and II showed substantial activation on 25% PS compared with 10% PS GUVs (*Figure 4D and E*). As shown before, the activity of complex II on 10% PS (DO base lipids) was lower than that of complex I (*Figures 1D* and *4F*). However, when the two complexes were compared on 25% PS (DO high PS), complexes I and II had nearly the same activity, suggesting a higher preference of complex II for negative charge (*Figure 4G*).

## Membranes influence structures and dynamics of the VPS34 complexes

We used hydrogen-deuterium exchange mass-spectrometry (HDX-MS) to determine the regions in the complexes I and II involved in membrane interaction. The advantage of HDX-MS is that it can map direct interactions of the native, unlabelled protein with membranes as well as mapping conformational changes associated with membrane binding (*Masson et al., 2016*).

The HDX-MS experiment for complex I had a sequence coverage of 87% (*Supplementary file 4*). Several regions in complex I showed a reduction in solvent exchange in the presence of 100 nm LUVs (*Figure 5*). The most pronounced HDX reduction was in a region of the BATS domain of ATG14L (residues 478–492), with additional changes in Beclin 1 and VPS34. In Beclin 1, three regions show protection in the presence of membranes, one in coiled-coil II (residues 201–212), another in the CC2/BARA linker (residues 262–274) and the third in a region that has been referred to as the BARA domain aromatic finger (AF1, residues 343–359 *Huang et al., 2012*). In the VPS34 kinase

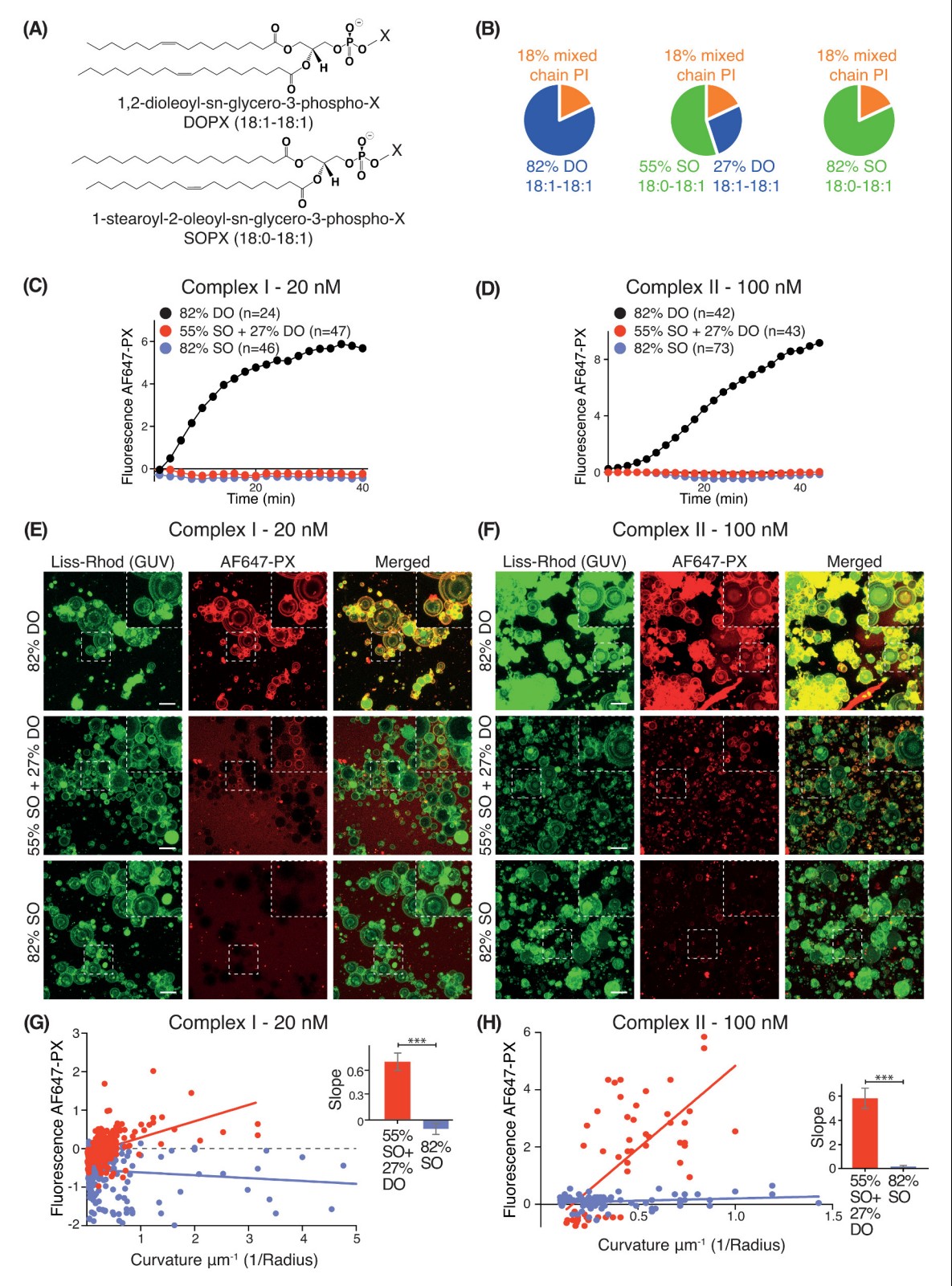

**Figure 2.** Effect of non-substrate lipid packing on complexes I and II. (**A**) Structures of a phospholipid with DO (1,2-dioleoyl) acyl chains or SO (1-stearoyl-2-oleoyl) acyl chains. (**B**) Lipid mixtures used to examine the effects of saturation of the non-substrate, 'background' lipids on activities of complexes I and II. The 82% DO lipids means that all of the background lipids (PC, PS and PE) are 1,2-dioleoyl (DO) lipids. Similarly, 82% SO means all of the background lipids are 1-stearoyl-2-oleoyl. (**C**) GUV assays for complex I using the lipid compositions shown in (**B**). Only the DO background lipids

*Figure 2 continued on next page*

*Figure 2 continued*

(82% DO) show significant activity. (D) GUV assays for complex II as in (C). (E) and (F) Z-stack analysis of (C) and (D). At 60 min, ten Z slices were obtained from an area, then projected into one plane with maximum projection to cover a wide range of sizes. Upper right cropped areas: Magnified images of the boxed areas. Scale bars: 15 µm. (G) and (H) Correlation between fluorescence intensity and membrane curvature (1/GUV Radius) from the same data as in (E) and (F), respectively. Right: Slopes for the plots. ***: p<0.001 (for (G) p<0.0001), (for (H) p<0.0001). For clarity, only mean values are plotted for each time point. Plots with SDs for each time point are in *Supplementary file 3*.

domain, helices k α1 and k α2 and the loop connecting them (residues 558–573) had a reduction in HDX. When the protected areas were overlaid onto a model of human complex I, the AF1 of Beclin 1, and the k α1-k α2 of VPS34 delineate a putative membrane-binding surface (*Figure 5B*). Although

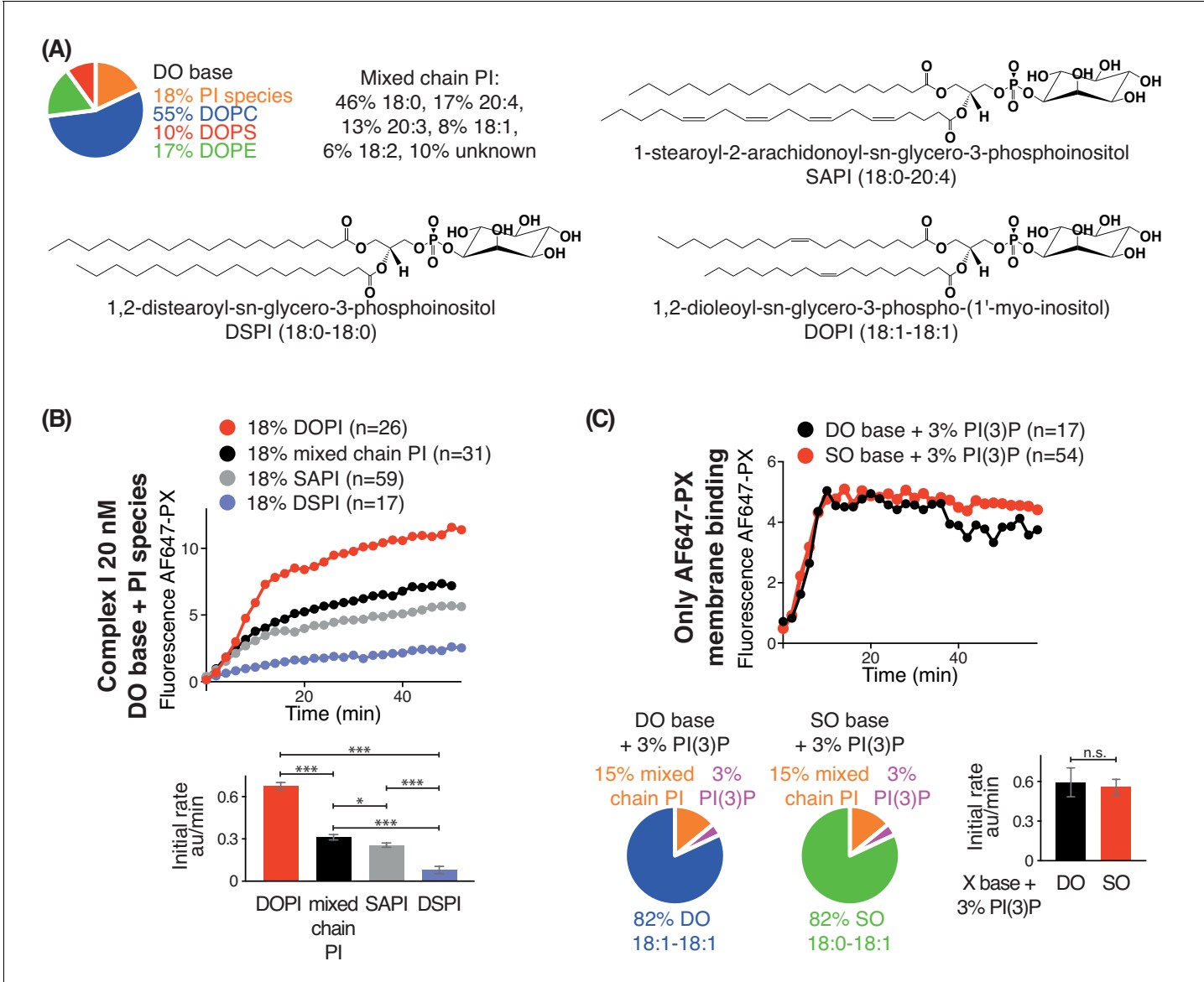

**Figure 3.** Increasing substrate PI acyl chain unsaturation increases complex I activity. (A) Structures of the PI lipids used in the context of the DO base background lipids. (B) Activities of complex I on GUVs containing the various PI substrates shown in (A) in the context of DO base background lipids. ***: p<0.001 (for all p<0.0001); *: p<0.05 (mixed chain PI vs SAPI p=0.0250). (C) Binding of the AlexaFluor 647-labelled p40 PX domain to 3% PI(3)P-containing GUVs is not dependent on lipid saturation of the background lipids. n.s.: not significant (p=0.7645). For clarity, only mean values are plotted for each time point. Plots with SDs for each time point are in *Supplementary file 3*.

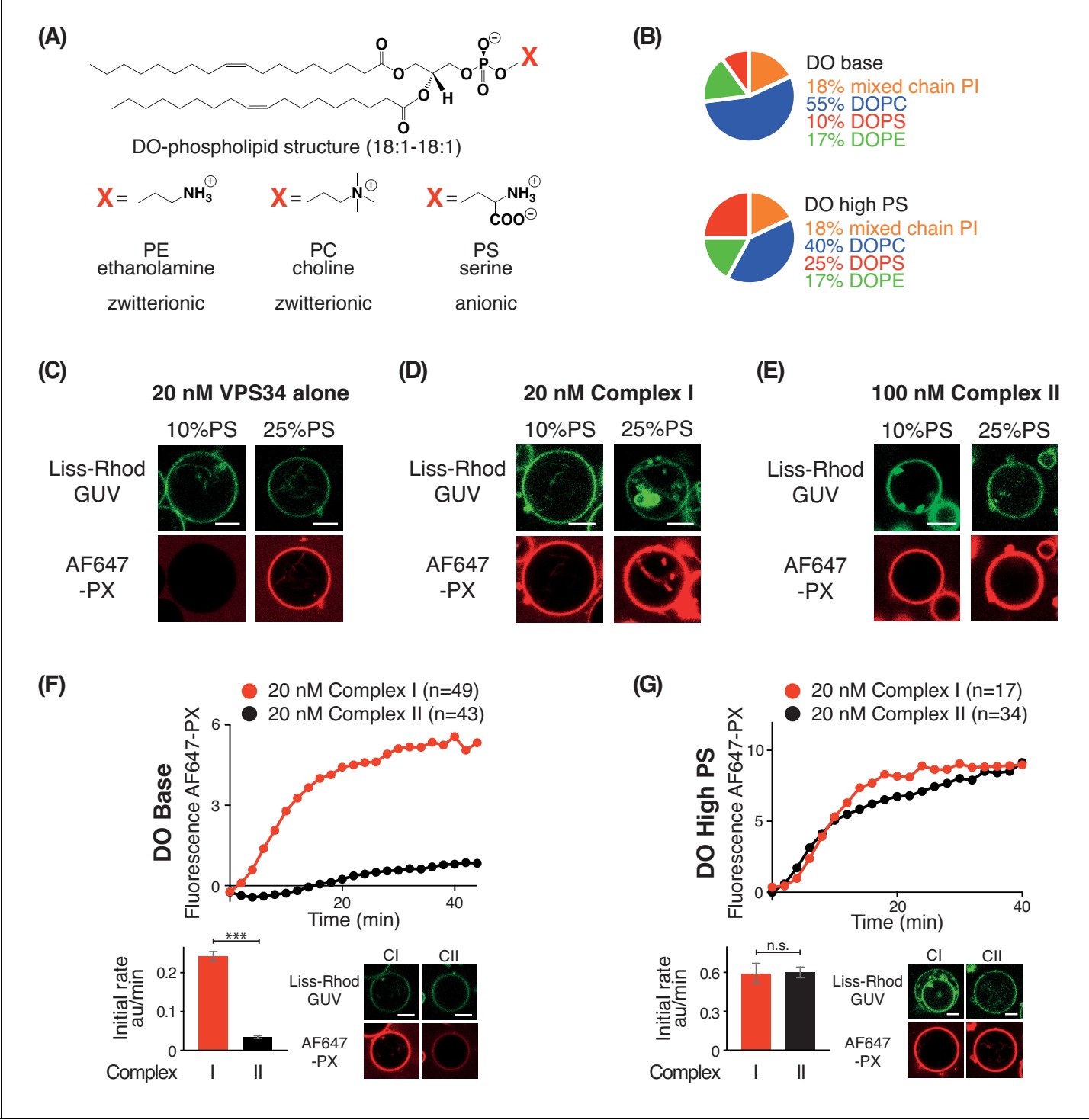

**Figure 4.** PS is an activator for complex I, complex II, and VPS34 alone. (A) Structures of DO phospholipid backbone (top) and head group for PE, PC, and PS (bottom). (B) Lipid compositions used to examine the influence of additional PS on VPS34 activity. (C–E) Activities of VPS34 alone, complex I and complex II on GUVs containing 10% PS (DO base) and 25% PS (DO high PS). Images were obtained 60 min after adding enzyme. (C) Activity of VPS34 alone (20 nM) on GUVs with 10% PS (DO base) and 25% PS (DO high PS). (D) Activity of complex I (20 nM) on GUVs with 10% PS (DO base) and 25% PS (DO high PS). (E) Activity of complex II (100 nM) on GUVs with 10% PS (DO base) and 25% PS (DO high PS). (F) and (G) Comparison of complexes I and II (20 nM) on GUVs with 10% PS (DO base) (F), or on 25% PS (DO high PS) (G). (F) ***: p<0.001 (p<0.0001); (G) n.s.: not significant (p=0.9274). For clarity, only mean values are plotted for each time point. Plots with SDs for each time point are in *Supplementary file 3*. All scale bars: 5 µm.

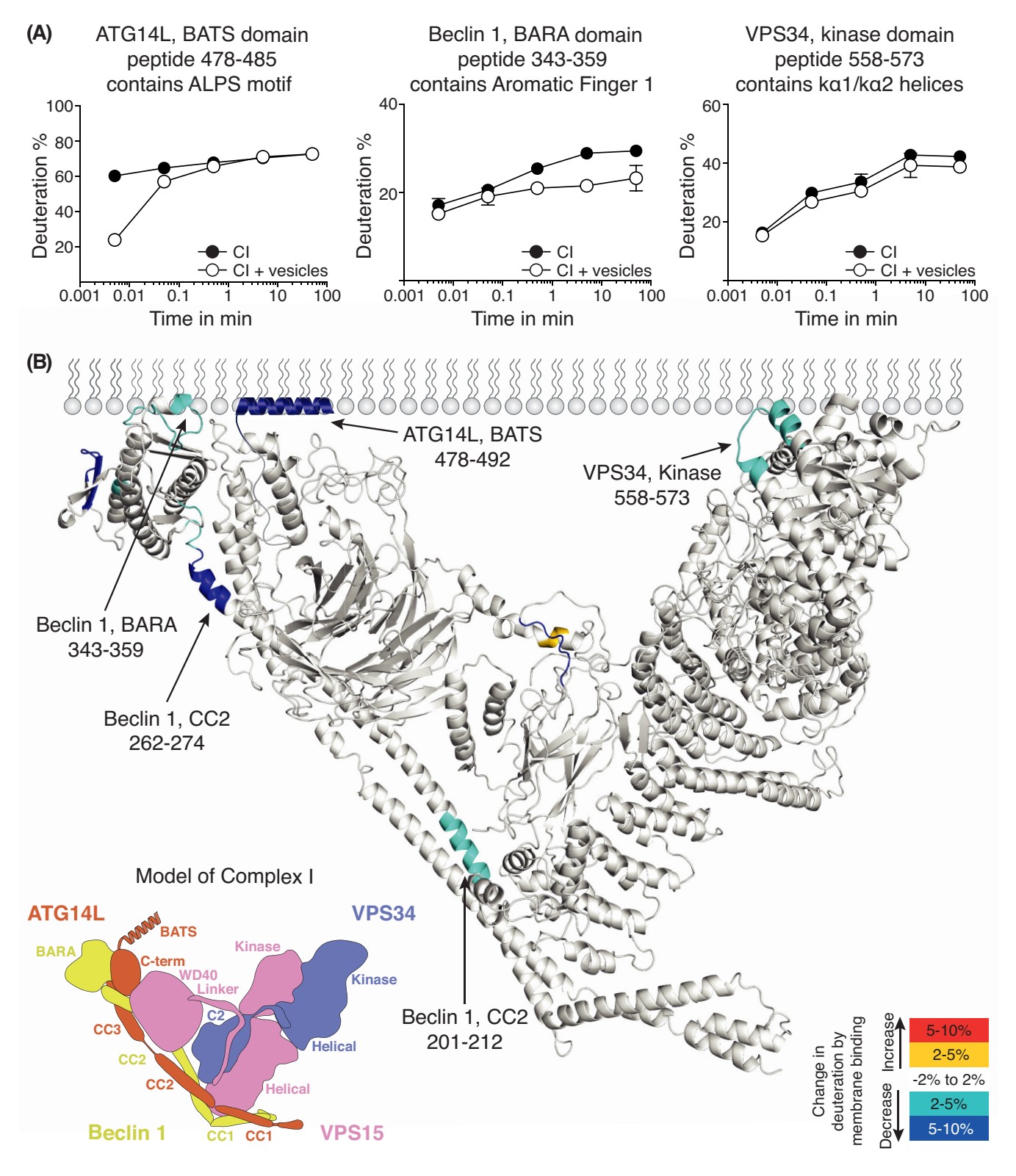

**Figure 5.** Analysis of membrane binding of human complex I using HDX-MS. (**A**) Representative peptides in complex I that showed significant HDX changes in the presence of lipids. (**B**) Overview of HDX changes illustrated on a model of human complex I. The model was built using SWISSMODEL and the structure of yeast complex II (PDB ID: 5DFZ). Peptides showing changes in HDX upon membrane binding are indicated by arrows. The model is colored by differences in HDX between the absence and presence of membranes (right inset). The left inset shows a schematic of complex I, identifying

*Figure 5 continued on next page*

Figure 5 continued

the subunits. CC1: coiled-coil I; CC2: coiled-coil II; CC3: coiled-coil III; BARA: β-α-repeated, autophagy-specific domain; C2: C2 domain; Helical: helical domain; Kinase: kinase domain; WD40: WD40 domain; BATS: BATS domain; C-term: ATG14L C-terminal domain.

there is no structure for the BATS domain, this region was shown to be involved in membrane binding (*Fan et al., 2011*).

We also performed HDX-MS for complex II in the presence and absence of 100 nm LUVs (*Figure 6* and *Supplementary file 5*). We see prominent protection in the presence of membranes in Beclin 1 and VPS34. The Beclin 1 CC2/BARA linker had decreased HDX in the presence of membranes in complex II similarly to complex I. Furthermore, as in complex I, the aromatic finger (AF1) of the Beclin 1 BARA domain showed decreased HDX in complex II (residues 358–359). Surprisingly, two additional regions, 288–300 and 409–419, were picked up in Beclin 1 at the putative membrane interface. We will refer to 288–300 as the hydrophobic loop (HL) and 409–419 as the aromatic finger 2 (AF2). In VPS34 (in the context of complex II), a peptide (874-884) from the C-terminal helix of VPS34 catalytic subunit showed protection. We had previously shown that the C-terminal helix is critical for catalysis on membranes (*Miller et al., 2010*). Although both complexes I and II show protection in the same domains, Beclin 1 BARA and VPS34 kinase, there are differences in the peptides protected that suggest that the two complexes assume similar but distinct orientations on membranes.

## The ATG14L BATS domain is critical for the activity of complex I on membranes

Although it has been shown that the BATS domain is important for membrane binding, it is not clear how membrane properties affect this interaction and complex I activity (*Brier et al., 2019*; *Fan et al., 2011*; *Ma et al., 2017*). Sequence analysis suggests that the BATS domain has an amphipathic α-helix referred to as the amphipathic lipid packing sensor (ALPS) (residues 471–492), and hydrophobic residues 484W, 485F, and 488Y within this motif are important for the binding of the BATS domain to SUVs and for the localization of ATG14L to autophagosomes (*Fan et al., 2011*). Using the GUV assay, we compared activities of complex I carrying a BATS mutation in which the ALPS motif was truncated (ΔALPS 471–492) with the wild-type complex I (WT) (*Figure 7A*). On DO base lipids (*Figure 7B*), the ΔALPS mutant showed no activity, suggesting that the motif is essential for the complex I activity on membranes. Even on GUVs made with DO high PS, the ΔALPS mutant of complex I showed hardly any detectable activity (*Figure 7C*). Surprisingly, complex I ΔALPS displayed significantly less activity than VPS34 alone on GUVs containing 25% PS (*Figure 7C*). This suggests that the non-catalytic subunits of complex I partially inhibit basal VPS34 activity in the absence of the ALPS motif. To determine whether the increase in activity conferred on complex I by the BATS domain is an intrinsic property of the domain, we fused the BATS domain to the C-terminus of the full-length UVRAG subunit and co-expressed it with other complex II subunits (*Figure 7A*). This fusion complex was shown previously to increase membrane binding of complex II (*Ma et al., 2017*). We found that the fusion complex II (UVRAG+BATS) showed increased activity, about 7-fold compared with the wild-type complex II (CII WT, *Figure 7D*). The human UVRAG has a 235 amino-acid long C-terminal extension compared to its yeast orthologue Vps38. When this UVRAG-specific extension was deleted and the BATS domain was fused to the truncated UVRAG, the resulting complex II variant (UVRAG ΔC+BATS, *Figure 7A*) showed an even higher activation of 11-fold over the WT complex II (*Figure 7D*). This suggests that the presence or absence of the UVRAG C-terminal region alters the orientation of the BATS domain with respect to the membrane. Interestingly, both UVRAG+BATS and UVRAG ΔC+BATS exhibited even higher activity than complex I WT (*Figure 7D*). Our activity results are consistent with the previous studies showing that the presence of the BATS domain increased membrane binding (*Ma et al., 2017*). Together with our HDX-MS results, this strongly indicates that the BATS domain activates complex I through a mechanism that involves an increase in membrane binding, and that the BATS domain can act as an autonomous membrane-interacting module that confers gain of function to VPS34 activity.

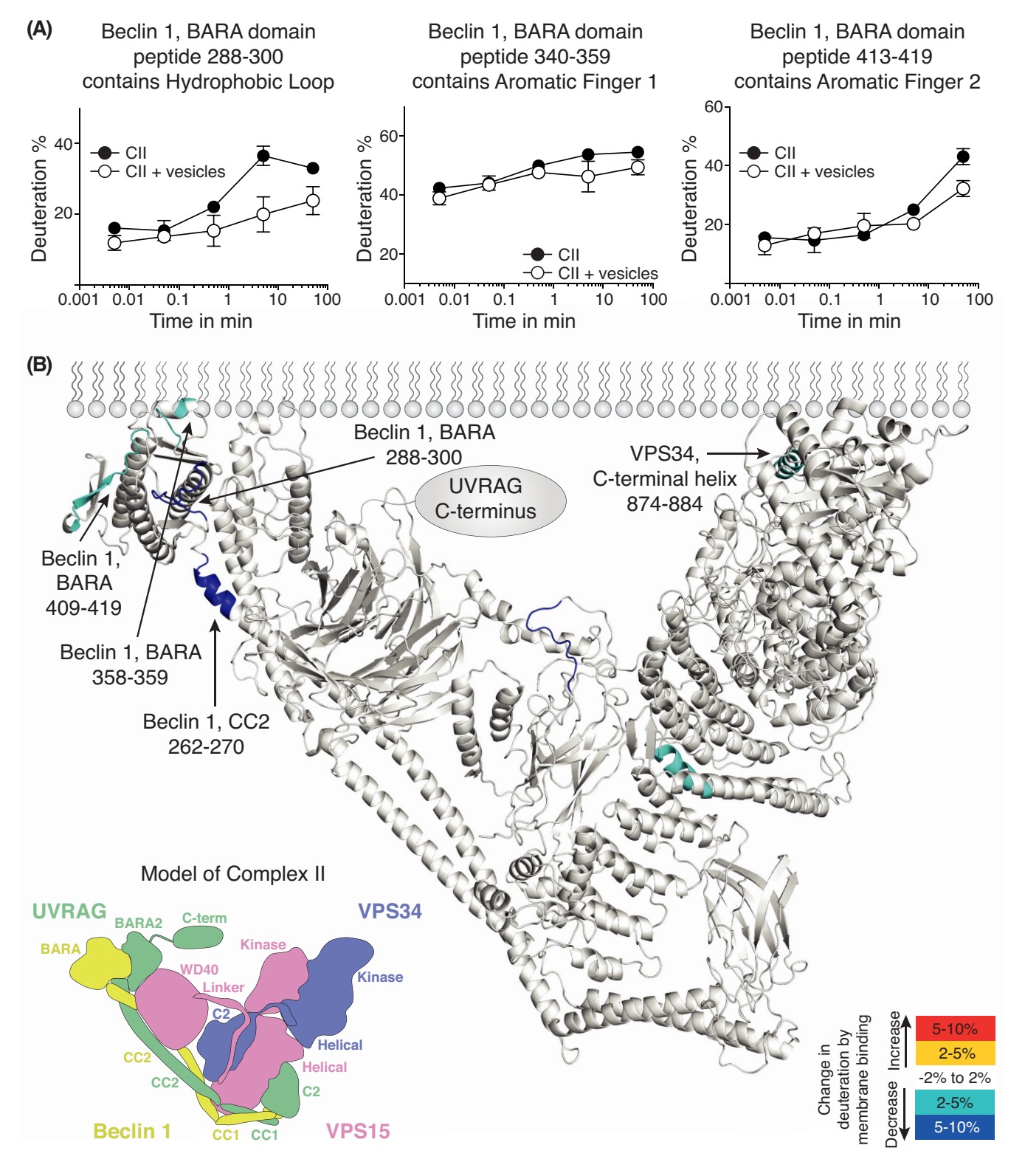

**Figure 6.** Analysis of membrane binding of human complex II using HDX-MS. (**A**) Representative peptides in complex II that showed significant HDX changes in the presence of lipids. (**B**) Overview of HDX changes illustrated on a model of human complex II. The model was built using SWISSMODEL and the structure of yeast complex II (PDB ID: 5DFZ). Peptides showing changes in HDX upon membrane binding are indicated by arrows. The model is colored by differences in HDX between the absence and presence of membranes (right inset). The left inset shows a schematic of complex II identifying

*Figure 6 continued on next page*

*Figure 6 continued*

the subunits and domains. CC1: coiled-coil I; CC2: coiled-coil II; BARA: β-α-repeated, autophagy-specific domain; BARA2: BARA2 domain; C2: C2 domain; Helical: helical domain; Kinase: kinase domain; WD40: WD40 domain; C-term: UVRAG C-terminal extension.

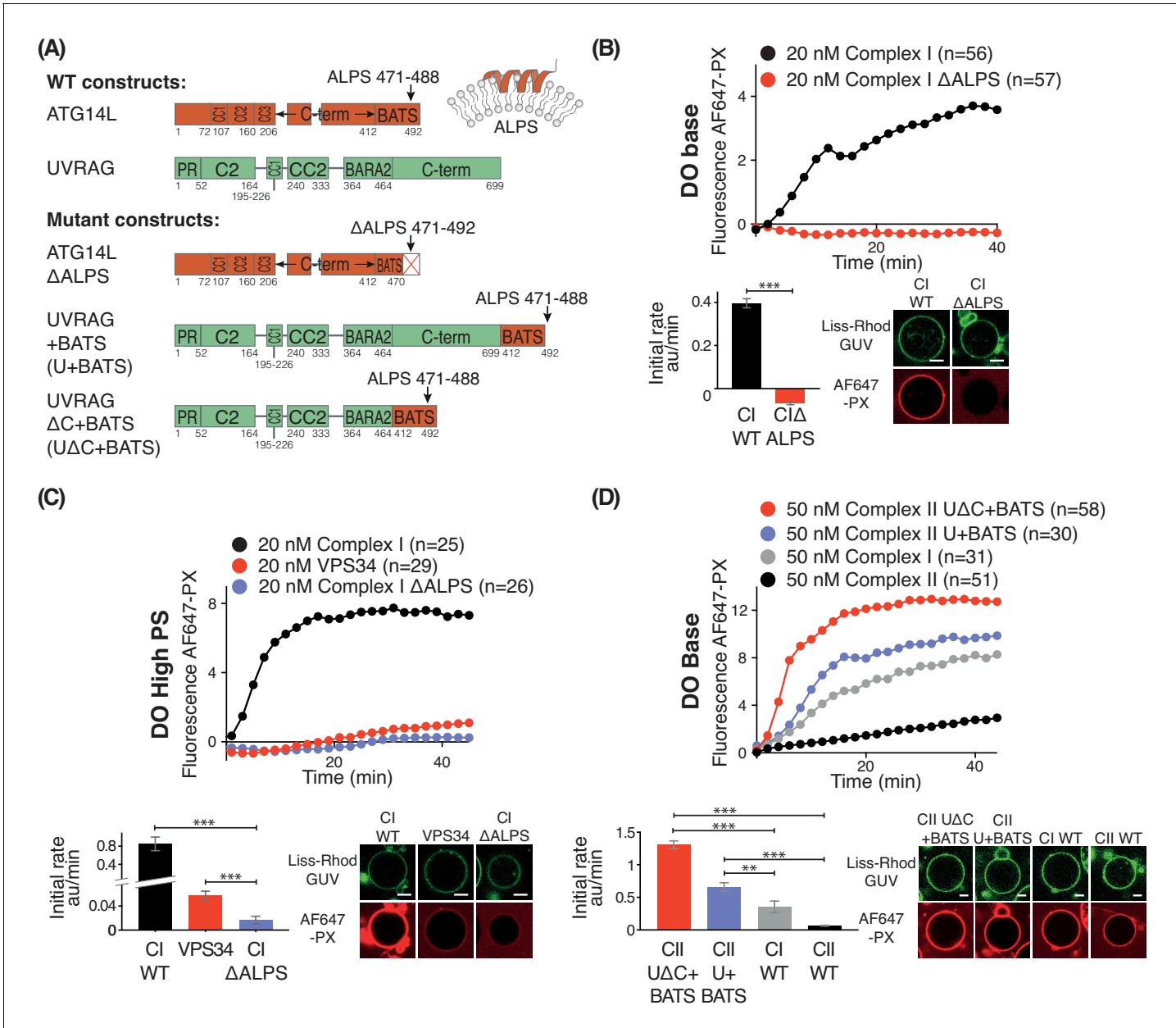

**Figure 7.** Influence of the Amphipathic Lipid Packing Sensor (ALPS) motif in the ATG14L BATS domain on the activities of complexes I and II. (**A**) Constructs used to examine the influence of the BATS domain on VPS34 activity in the contexts of complex I and a complex I/II chimera on GUVs. (**B**) The ALPS motif of the BATS domain is important for the activity of complex I. Upper: Activities of wild-type complex I (WT), and complex I with the ΔALPS mutations on GUVs with DO base lipids. Lower left: Initial rates. Lower right: Confocal micrographs of the GUV assay at 60 min after adding proteins. ***: p<0.001 (p<0.0001). (**C**) Activities of WT and ΔALPS complex I as well as the VPS34 kinase subunit on its own on GUVs with DO high PS. ***: p<0.001 (for all p<0.0001). (**D**) Activities on GUVs with DO base lipids for complex II and for fusions of the BATS domain to UVRAG C-terminus. U + BATS: UVRAG + BATS; UΔC + BATS: UVRAGΔC + BATS as illustrated in (**A**). ***: p<0.001 (for all p<0.0001); **: p<0.01 (CI WT vs CII U+BATS p=0.0059). For clarity, only mean values are plotted for each time point. Plots with SDs for each time point are in *Supplementary file 3*. All scale bars: 5 μm.

## The Beclin 1 BARA domain is more important for the activity of complex II than complex I

Three motifs, aromatic finger 1 (AF1, 359-FFW-361), aromatic finger 2 (AF2, 418-QF-419), and the hydrophobic loop motif (HL, 293-LPSVPV-298) found in the BARA domain of Beclin 1 are part of the putative membrane interface (*Figure 8A*). HDX-MS showed that membrane binding decreased HDX for only AF1 in complex I, whereas HL, AF1 and AF2 had reduced HDX in complex II (*Figure 8A*). This suggests that the BARA domain might bind membranes differently in the context of complex I and complex II (*Figure 8A*, middle panel). We measured the activities of both complexes I and II when carrying mutations in these motifs (AFM1 = F359D F360D W360D, AFM2 = Q418D F419D, and HLM = L293D V296D) in order to examine their importance. This revealed that complex I carrying AFM1 or AFM1+HLM (both AFM1 and HLM) had reduced but still significant activities (*Figure 8B*), while AFM2 did not affect the complex I activity (*Figure 8C*). In contrast, complex II activity depended on all three motifs (*Figure 8D*). This suggests that complex I activity is more dependent on the BATS domain than the BARA domain, with HL and AF2 dispensable for complex I. On the other hand, complex II activity is critically dependent on the BARA domain, where AF1, AF2 and HL are all essential for full activity.

## Phosphoinositides have modest effects on activities of complexes I and II

Phosphoinositides (PIPs) represent about 10% of total phospholipids, and less than 1% of total lipids in biological membranes (*De Craene et al., 2017*; *Payrastre et al., 2001*). However, specific PIPs are enriched in distinct organelle membranes, and this helps to selectively recruit proteins. Since previous work established the importance of kinases that generate phosphatidylinositol 4-phosphate (PI(4)P) and phosphatidylinositol 4,5-bisphosphate (PI(4,5)$P_2$) in autophagy (*Wang et al., 2015*; *Tan et al., 2016*), we examined the influence of PI(4)P and PI(4,5)$P_2$ on complexes I and II. For this, we added either 5% DOPI(4)P or 5% DOPI(4,5)$P_2$ in place of 5% DOPC to the DO base lipids (*Figure 9A, B and C*). We found that complex I was activated by PI(4)P (~1.8 fold) and by PI(4,5)$P_2$ (~1.2 fold) compared to DO base lipids (*Figure 9D and E*). In contrast, complex II was also modestly activated by PI(4)P (~1.7 fold) (*Figure 9F*) but unaffected by PI(4,5)$P_2$ (*Figure 9G*). The activation of complex I by PI(4,5)$P_2$ is consistent with the observation that the isolated BATS domain of ATG14L has a binding preference for PI(4,5)$P_2$, in addition to its ability to sense membrane curvature (*Fan et al., 2011*; *Tan et al., 2016*).

## Discussion

In this study, we systematically examined the importance of three physicochemical membrane parameters on the lipid kinase activities, structures and dynamics of VPS34 complexes I and II. We found that all three parameters, packing defects caused by acyl chain unsaturation for both substrate and non-substrate lipids, membrane curvature and electrostatics, significantly affected the activity of both complexes. It was surprising that changing the saturation status on just one of the two acyl chains dramatically altered the activity of both complexes. Variations in acyl chain composition of glycerophospholipids can give rise to thousands of lipid species. Nevertheless, mammalian cells have some distinct patterns of acyl chain enrichments. One notable example of this is that 40–85% of PI is present as SAPI (18:0/20:4) (*Barneda et al., 2019*; *Blunsom and Cockcroft, 2020*; *Bozelli and Epand, 2019*). We showed that VPS34 kinase is activated by unsaturation of its substrate PI (*Figure 3B*). This is consistent with the observation in *C. elegans* that depletion of an enzyme that incorporates polyunsaturated fatty acids (PUFAs) in the *sn-2* position of PI significantly decreased PI (3)P content and size of early endosomes, without reducing total PI (*Lee et al., 2008*; *Lee et al., 2012*). More surprising was our result that increasing unsaturation of the background lipids also dramatically increased activities of complexes I and II (*Figure 2*). The effect of background lipids unsaturation is likely to be mediated largely by the adaptor arm, since VPS34 alone shows no activity on DO base (10% PS) lipids (*Figure 4C*), whereas complexes I and II on the same lipid mixture are active (*Figure 4D and E*). Stearoyl-CoA desaturase activity is important for early steps in autophagy (*Ogasawara et al., 2014*; *Köhler et al., 2009*), and recent lipidomic analyses have shown that autophagic membranes of budding yeast contain as much as 60% lipids with two unsaturated fatty acyl

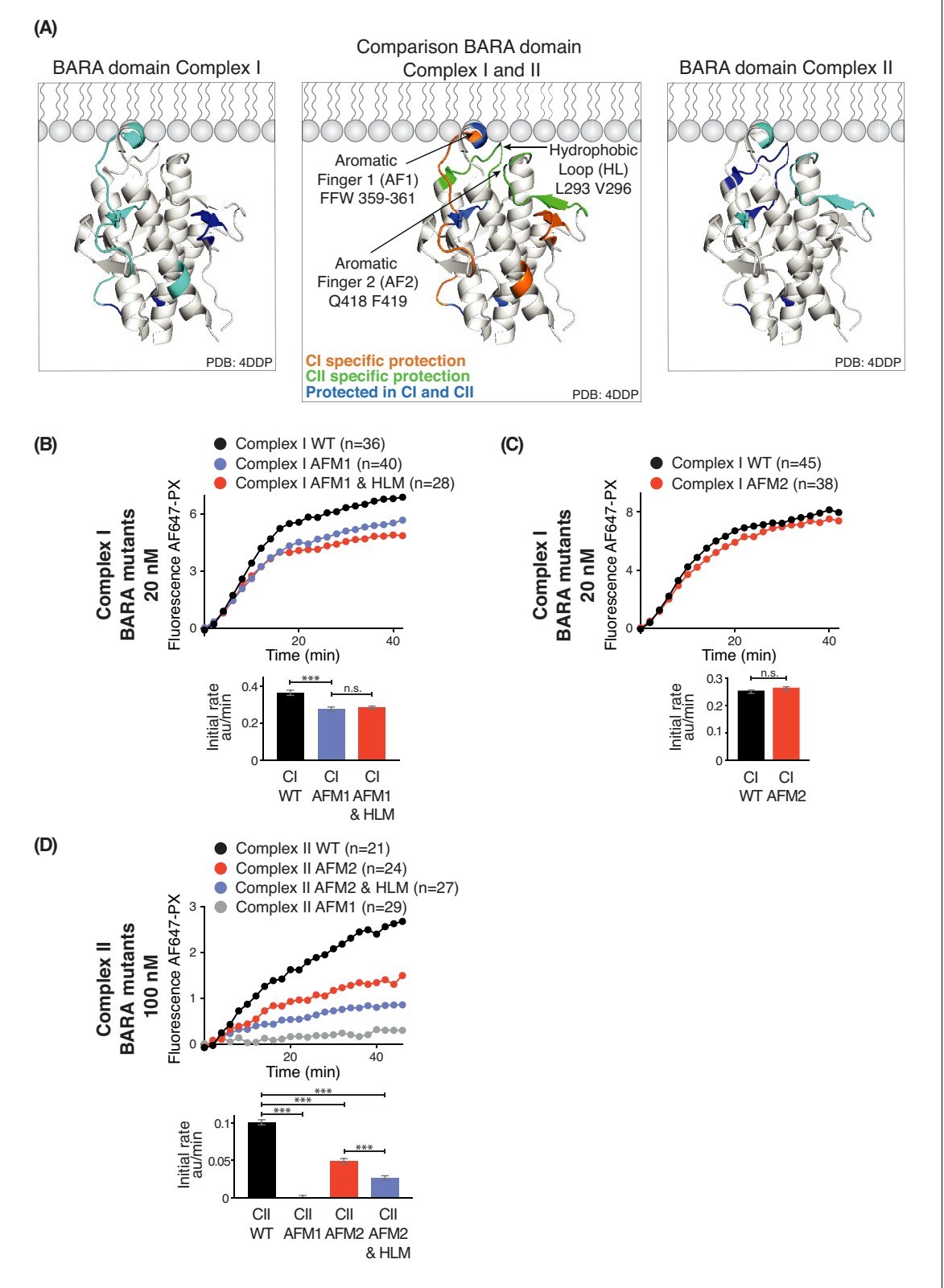

**Figure 8.** Influence of Beclin 1 BARA domain mutations on the activities of complexes I and II. (**A**) Summary of HDX changes for the Beclin 1 BARA domain of human complex I (left) and II (right) overlaid on a ribbon diagram of the structure of the domain (PDB ID 4DDP). In the middle panel, peptides are coloured by whether they are specifically protected in complex I (orange), complex II (green), or both complexes I and II (blue). Elements involved in HDX changes are AF1 (Aromatic finger motif 1; 359-FFW-361), AF2 (Aromatic finger motif 2; 418-QF-419), and HL (Hydrophobic loop; 293-

*Figure 8 continued on next page*

Figure 8 continued

LPSVPV-298). (**B–D**) Effects of mutations in the membrane-binding elements of complexes I and II on activities using GUVs with DO base lipids. Top: Reaction progress curves. Only mean values are plotted. Plots with SDs are in *Supplementary file 3*. Bottom: initial rates. (**B**) AFM1 and HL mutations in the BARA domain modestly affect the activity of complex I on GUVs. Beclin 1 constructs: wild-type (WT), aromatic finger one mutant (AFM1), and AFM1+ hydrophobic loop mutant (AFM1+HLM). ***: p<0.001 (p<0.0001); n.s.: not significant (p=0.4238). (**C**) AFM2 mutation has no influence on complex I activity. n.s.: not significant (p=0.5186). (**D**) All of the mutations have an effect on complex II activity. ***: p<0.001 (for all p<0.0001). For clarity, only mean values are plotted for each time point. Plots with SDs for each time point are in *Supplementary file 3*. ***: p<0.001; n.s.: not significant.

moieties, and only 2% with no double bonds (*Schütter et al., 2020*). The potent effect of lipid unsaturation on VPS34 complexes combined with the compartment-specific lipid unsaturation suggests an important means of modulating VPS34 activity on cellular organelles. The importance of lipid unsaturation for VPS34 activity is analogous to what was reported for another lipid kinase involved in the generating polyphosphoinositides. Increasing acyl chain unsaturation for substrate and non-substrate lipids in detergent micelles activates phosphatidylinositol-4-phosphate 5-kinase isoforms (*Shulga et al., 2012*).

The human VPS34 complexes were activated by increased membrane curvature (*Figure 2*), which is consistent with results for the yeast complexes I and II (*Rostislavleva et al., 2015*) and human complexes I and II (*Brier et al., 2019*). An alternative possibility that cannot be excluded is that smaller GUVs incorporate more unsaturated lipid during their formation and hence are able to recruit the two complexes more efficiently. Membrane curvature causes tilting of lipids and decreased lipid packing, thereby opening hydrophobic cavities that can be sensed by protein motifs like the ALPS motif from complex I (*Vanni et al., 2014*) and motifs from the BARA domain of both complexes I and II. Although the VPS34 complexes make protein/protein interactions that help localize them to specific lipid compartments, the preference for high curvature and lipid unsaturation may contribute to the lack of activity of these PI3K complexes at the PM, which is enriched with saturated lipids and has a lower curvature (*Bigay and Antonny, 2012*). Rab5 is a primary determinant for recruiting complex II to early endosomes. Interestingly, like complex II, Rab5 has a preference for target membranes with high curvature and monounsaturated lipids, and this preference derives from the unsaturated prenyl lipids attached to Rab5 fitting into packing defects in the target membrane (*Kulakowski et al., 2018*). Curvature and lipid packing are closely linked, consequently, the loss of VPS34 activity caused by tight lipid packing of saturated lipids can be counteracted by an increase in membrane curvature (*Figure 2*).

As the third physicochemical parameter, we tested the effects of electrostatics on the activities of VPS34 complexes by increasing the concentration of phosphatidylserine (PS). PS is enriched on the cytosolic leaflet of the plasma membrane (PM) and endocytic vesicles mainly at recycling endosomes (RE) (*Fairn et al., 2011*). The PS on the cytosolic leaflet plays important roles in the regulation of endocytic pathways (*Uchida et al., 2011*). We found that PS activates complexes I and II, and also VPS34 alone (*Figure 4C–E*). Significantly, GUVs with 25% PS enabled complex II to achieve the same activity as complex I (*Figure 4G*), whereas on membranes with 10% PS, complex I was much more active than complex II (*Figure 4F*). The ability of high PS content to strongly activate complex II might contribute to complex II activity on early endosomes, which have more PS than the ER (*Hullin-Matsuda et al., 2014*). It is reasonable to hypothesize that PS, together with Rab5, may contribute to activation of complex II on early endosomes. Increased PS in combination with high curvature and loosely packed lipids may contribute also to activation of complex I during autophagy. In cells, PS is synthesized by PS synthases (PSS) I and II at the ER-mitochondria-associated membranes (MAM) (*Lagace and Ridgway, 2013*). Upon starvation, complex I is recruited to the MAM (*Hamasaki et al., 2013*). The PSS I co-localizes with FIP200, a member of autophagy-initiation complex, which is necessary for the recruitment of complex I (*Nishimura et al., 2017*). ER membranes are enriched in loosely packed lipids, and the site of initiation of the phagophore is a highly curved structure known as an omegasome (*Axe et al., 2008*). This would provide an ideal platform to activate complex I. However, membrane properties alone would not explain the proclivity of complex I to provide PI3K activity in autophagy compared with complex II. Additional interactions could contribute to enrichment of complex I on autophagosomes. Recently, it was shown that the ATG14L BATS domain may be acting as a coincidence detector, using its ALPS motif to interact with membranes and its LIR

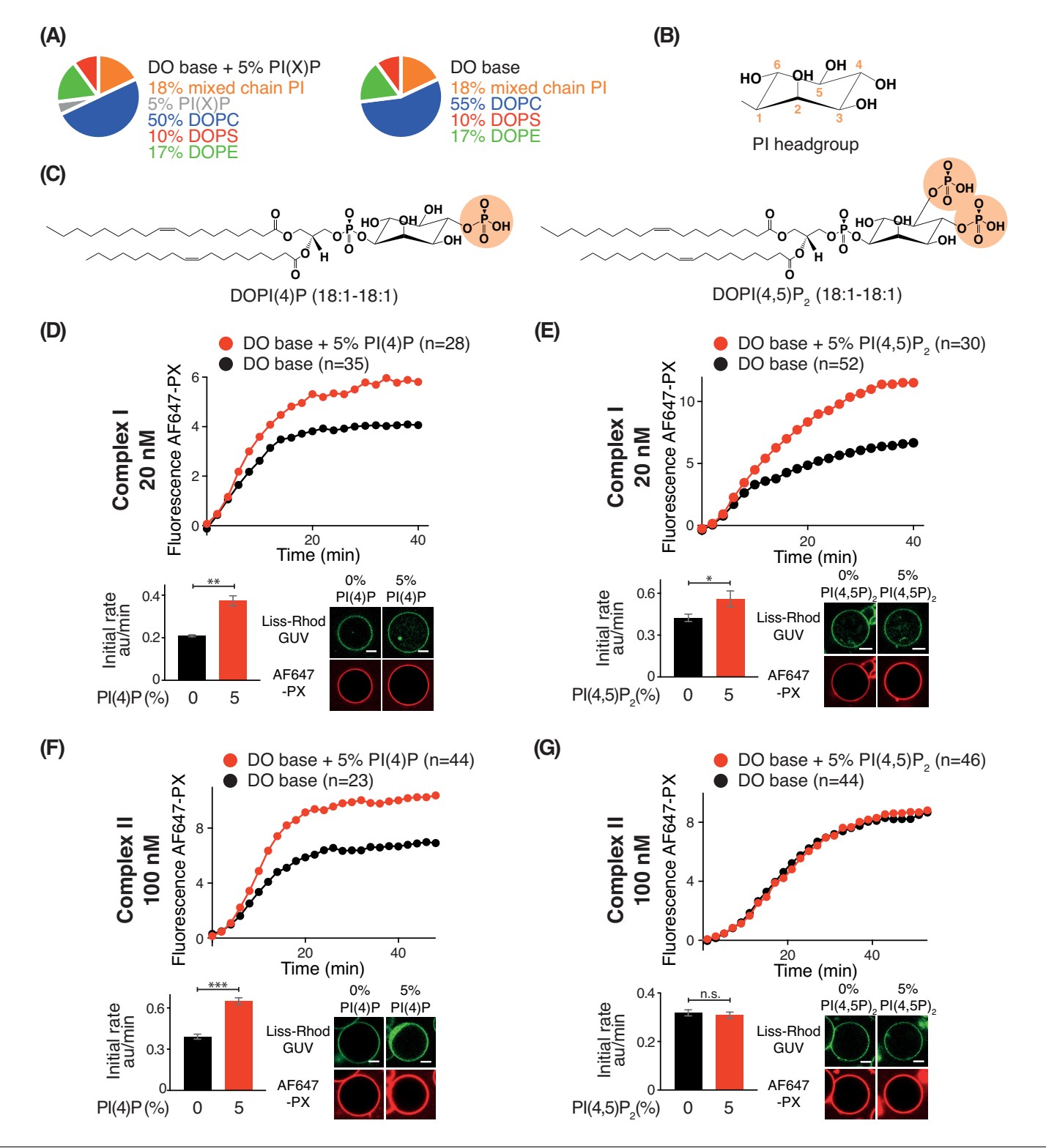

**Figure 9.** Effects of phosphoinositides on complexes I and II. Activities of GUVs in which 5% of the PC in the base mixture is replaced by either 5% PI(4)P or 5% PI(4,5)P$_2$ compared with base lipid-containing GUVs. (**A**) Composition of GUVs. (**B**) General structure of a phosphatidylinositol headgroup. (**C**) Structures of DOPI(4)P and DOPI(4,5)P$_2$. (**D-G**) For each figure, the upper panel shows the reaction time course, the lower left panel shows the initial rates and the lower right illustrates confocal micrographs 60 min after adding the enzyme complexes. (**D**) Influence of 5% DOPI(4)P on complex I activity. **: $p < 0.01$ ($p = 0.0059$). (**E**) Influence of 5% DOPI(4,5)P$_2$ on complex I activity. *: $p < 0.05$ $p = 0.0140$). (**F**) Influence of 5% DOPI(4)P on complex II

*Figure 9 continued on next page*

*Figure 9 continued*

activity. ***: p<0.001 (p<0.0001). (**G**) Influence of 5% DOPI(4,5)P$_2$ on complex II activity. n.s.: not significant (p=0.5796). For clarity, only mean values are plotted for each time point. Plots with SDs for each time point are in ***Supplementary file 3***. All scale bars: 5 µm.

motif to interact with autophagic GAPARAPs (*Birgisdottir et al., 2019*). Along with the ULK1 complex, they may contribute to autophagosome-specific localization and activation.

Both the HDX-MS results and the mutational analyses suggest that complexes I and II have some complex-specific as well as common interactions with membranes. On GUVs made with DO base lipids, human complex I is considerably more active than complex II, which stands in contrast to our results for yeast VPS34 complexes, where complex II was more active than complex I (*Rostislavleva et al., 2015*). The contrasting behaviour of human and yeast VPS34 complexes likely arises from extensive sequence differences between human and yeast Atg14, particularly in the region corresponding to the C-terminal BATS of ATG14L. The complex I BATS domain interaction with membranes is the most striking of the complex-specific membrane interactions in the human VPS34 complexes. The BATS dominates the complex I membrane interaction. When the BATS domain is deleted, almost all of complex I activity is eliminated (*Figure 7*). Furthermore, complex I binds to membranes more strongly than complex II (*Figure 1D* and *Ma et al., 2017*). Adding the BATS domain to either full-length UVRAG or a UVRAG fragment lacking the C-terminus dramatically increased complex II activity (*Figure 7D*) and membrane binding *in vitro* and in cells (*Ma et al., 2017*). Our results suggest that the differences in activities for complexes I and II reflect a lower affinity of complex II for lipid vesicles (*Figure 1C and D*). In cells, this weak membrane affinity of complex II may be compensated by its binding partners, such as Rab5, which help to recruit complex II to membranes (*Christoforidis et al., 1999*). Our results also show that the activities of complexes I and II become equalized on lipid vesicles with increased PS, and this is consistent with a previous report that complexes I and II have identical activities on LUVs formed with 50%PS and 50%PI (*Brier et al., 2019*).

To our surprise, the ALPS deletion mutant showed even lower activity than VPS34 alone (*Figure 7C*), suggesting that the influence of the ATG14L subunit is intrinsically inhibitory, but that this inhibition is overcome by the much stronger membrane binding conferred by the BATS domain, resulting in an overall gain of function. A similar mechanism has been reported for the H-Ras-PI3Kα complex, where H-Ras binding is inhibitory for PI3Kα, but the inhibition is overcome by increased recruitment of PI3Kα to membranes (*Buckles et al., 2017*). The presence of the ATG14L subunit alters the interactions of the rest of the complex with membranes. This subunit appears to enhance the association of the VPS34 kinase domain with membranes (giving rise to protection of the kα1-kα2 region of VPS34; *Figure 5*). It also alters the interaction of Beclin 1 with membranes, so that while the BARA domain AF1 motif has a role in membrane binding for both complexes, AF2 and HL motifs are important only for complex II (*Figures 5*, *6* and *8*). However, in the context of the dominant role of the BATS domain on complex I activity, the influence of the Beclin 1 AF1 on activity is minor, and there is no significant influence of AF2 or HL (*Figure 8*). This is consistent with the previous report that mutation of AF1 only mildly affects the autophagosome generation (*Huang et al., 2012*). It may be that the BATS domain affects the orientation of complex I, so that AF2 and HL no longer interact with membranes. In contrast to complex I, complex II depends critically on AF1, AF2 and HL to enable its maximal activity on membranes (*Figure 8*). The yeast complex II activity also depends on AF1 (*Rostislavleva et al., 2015*), indicating that the activation mechanisms of these two orthologous complexes are similar. The HL motif in the BARA domain (*Figures 6* and *8*) has S295 that can be phosphorylated by Akt (*Wang et al., 2012*), which enhances interactions of Beclin 1 with 14-3-3 and vimentin intermediate filament proteins. On the other hand, the S295A mutation enhances autophagy (*Wang et al., 2012*). In cells, the S295 phosphorylation by Akt and/or interaction with 14-3-3 may negatively affect membrane binding of complexes I and II.

In addition to the AF1 motif in the Beclin 1 BARA domain, another region showing decreased HDX in both complexes I and II is the CC2/BARA linker of Beclin 1 (residues 262–274 in complex I and residues 262–270 in complex II). It was proposed that hydrophobic residues Phe270 and Phe274 in this linker interact with lipid membranes (*Chang et al., 2019*). Consistent with this, mutation of Phe270 and Phe274 eliminated membrane binding and activity of complex II (*Chang et al., 2019*). The somewhat distant position of these two residues from the putative membrane-binding surface

marked by AF1, prompted the proposal that the adjacent BARA β1 and β2 strands become unlatched to allow Phe270/Phe274 to reach the membrane. Although these two hydrophobic residues were protected from HDX in the presence of lipid membranes in our studies (*Figures 5* and *6*), the peptides covering the β1-β2 of the BARA domain β-sheet 1 (residues 275–287, 276–294, *Supplementary files 4* and *5*) did not show significant HDX changes. In addition, in yeast, membranes cause this region to increase in HDX (residues 312–327 in Vps30) (*Rostislavleva et al., 2015*). Consequently, the question whether the CC2/BARA linker becomes more protected due to direct interaction with membranes or due to an indirect ordering in the presence of membranes remains an open question.

Both complexes I and II are activated by PI(4)P, This phosphoinositide is enriched on the Golgi apparatus (*Hammond and Balla, 2015*), early endosomes (*Henmi et al., 2016*), and autophagosomes (*Judith et al., 2019*; *Wang et al., 2015*), where complexes I and II are activated. The coincidence of Rab5, PS, and PI(4)P at early endosomes might help to specifically enhance complex II activity on this compartment. (*Judith et al., 2019*; *Munson et al., 2015*; *Sridhar et al., 2013*; *Wang et al., 2015*) We also found that PI(4,5)$P_2$ modestly increases activity of complex I (*Figure 9E*). This could be mediated by binding of PI(4,5)$P_2$ to the ATG14L BATS domain (*Wang et al., 2015*; *Tan et al., 2016*). Interestingly, it was recently shown that PIPKIγi5, which phosphorylates PI(4)P to produce PI(4,5)$P_2$, binds to ATG14L and localizes to autophagosomes (*Tan et al., 2016*). Compared with the effects of the three membrane physicochemical parameters, phosphoinositides PI(4)P and PI(4,5)$P_2$ have only modest effects *in vitro*.

Overall, our results uncovered the importance of lipid packing dictated by acyl chain unsaturation of both substrate and background lipids, negative charge, and membrane curvature for the activation of both complexes I and II. However, these complexes have unique cellular localization and HDX-MS analysis revealed differences in membrane binding modes between complexes I and II that are likely to contribute to their different roles.

# Materials and methods

## Key resources table

| Reagent type (species) or resource | Designation | Source or reference | Identifiers | Additional information |
|---|---|---|---|---|
| Chemical compound, drug | DOPC | Avanti Polar Lipids, Inc | 850375C | Dissolved in chloroform |
| Chemical compound, drug | DOPE | Avanti Polar Lipids, Inc | 850725C | Dissolved in chloroform |
| Chemical compound, drug | DOPS | Avanti Polar Lipids, Inc | 840035C | Dissolved in chloroform |
| Chemical compound, drug | Brain PC (Porcine) | Avanti Polar Lipids, Inc | 840053C | Dissolved in chloroform |
| Chemical compound, drug | Brain PE (Porcine) | Avanti Polar Lipids, Inc | 840022C | Dissolved in chloroform |
| Chemical compound, drug | Brain PS (Porcine) | Avanti Polar Lipids, Inc | 840032C | Dissolved in chloroform |
| Chemical compound, drug | Liver PI (mixed chain PI, Bovine) | Avanti Polar Lipids, Inc | 840042C | Dissolved in chloroform |
| Chemical compound, drug | DSPE-PEG(2000) Biotin | Avanti Polar Lipids, Inc | 880129C | Dissolved in chloroform |
| Chemical compound, drug | DO Liss Rhod PE | Avanti Polar Lipids, Inc | 810150P | Dissolved in chloroform |
| Chemical compound, drug | SOPC | Avanti Polar Lipids, Inc | 850467C | Dissolved in chloroform |
| Chemical compound, drug | SOPE | Avanti Polar Lipids, Inc | 850758C | Dissolved in chloroform |
| Chemical compound, drug | SOPS | Avanti Polar Lipids, Inc | 840039C | Dissolved in chloroform |

*Continued on next page*

*Continued*

| Reagent type (species) or resource | Designation | Source or reference | Identifiers | Additional information |
|---|---|---|---|---|
| Chemical compound, drug | DSPI | Avanti Polar Lipids, Inc | 850143P | Dissolved in chloroform: methanol:H2O = 20:9:1 |
| Chemical compound, drug | SAPI | Avanti Polar Lipids, Inc | 850144P | Dissolved in chloroform: methanol:H2O = 20:9:1 |
| Chemical compound, drug | DOPI | Avanti Polar Lipids, Inc | 850149P | Dissolved in chloroform: methanol:H2O = 20:9:1 |
| Chemical compound, drug | DOPI(3)P | Avanti Polar Lipids, Inc | 850150P | Dissolved in chloroform |
| Chemical compound, drug | DOPI(4)P | Avanti Polar Lipids, Inc | 850151P | Dissolved in chloroform |
| Chemical compound, drug | DOPI(4,5)P2 | Avanti Polar Lipids, Inc | 850155P | Dissolved in chloroform |
| Chemical compound, drug | Cholesterol | Sigma | C2044 | Dissolved in chloroform |
| Chemical compound, drug | 18:1 Sphingomyelin | Avanti Polar Lipids, Inc | 860587 c | Dissolved in chloroform |
| Recombinant DNA reagent | HsVPS34 + HsVPS15-3xTEV-ZZ | (*Ohashi et al., 2016*) | pYO1025 | See *Supplementary file 6* for more details |
| Recombinant DNA reagent | HsBeclin 1, untagged | This work | pYO1006 | See *Supplementary file 6* for more details |
| Recombinant DNA reagent | ZZ-3XTEV-HsATG14L | This work | pYO1017 | See *Supplementary file 6* for more details |
| Recombinant DNA reagent | ZZ-3XTEV-HsUVRAG | This work | pYO1018 | See *Supplementary file 6* for more details |
| Recombinant DNA reagent | HsBeclin 1 + ZZ-3XTEV-UVRAG | This work | pYO1023 | See *Supplementary file 6* for more details |
| Recombinant DNA reagent | HsBeclin 1 + HsUVRAG, both untagged | This work | pYO1031 | See *Supplementary file 6* for more details |
| Recombinant DNA reagent | HsBeclin 1 AFM1 (FFW/DDD at 359–361), untagged + ZZ-3xTEV-HsATG14L | This work | pYO1051 | See *Supplementary file 6* for more details |
| Recombinant DNA reagent | HsBeclin 1 AFM1 (FFW/DDD at 359–361), untagged + ZZ-3x TEV-HsUVRAG | This work | pYO1052 | See *Supplementary file 6* for more details |
| Recombinant DNA reagent | ZZ_3xTEV-ATG14 delta ALPS (1-470) | This work | pYO1077 | See *Supplementary file 6* for more details |
| Recombinant DNA reagent | HsBeclin 1 + HsATG14L, both untagged | This work | pYO1101 | See *Supplementary file 6* for more details |
| Recombinant DNA reagent | HsBeclin 1 AFM2 (Q418D F419D), untagged + ZZ-3xTEV-HsATG14L | This work | pYO1118 | See *Supplementary file 6* for more details |
| Recombinant DNA reagent | HsBeclin 1 AFM2 (Q418D F419D), untagged + ZZ-3xTEV-UVRAG | This work | pYO1120 | See *Supplementary file 6* for more details |
| Recombinant DNA reagent | HsUVRAG + HsATG14L BATS domain(413-492) fusion, untagged | This work | pYO1123 | See *Supplementary file 6* for more details |
| Recombinant DNA reagent | HsUVRAG delta Cter (1-464) + HsATG14L BATS domain(413-492) fusion, untagged | This work | pYO1124 | See *Supplementary file 6* for more details |

*Continued on next page*

*Continued*

| Reagent type (species) or resource | Designation | Source or reference | Identifiers | Additional information |
|---|---|---|---|---|
| Recombinant DNA reagent | GST-TEV-Cys-PX (2–149) | This work | pYO1125 | See *Supplementary file 6* for more details |
| Recombinant DNA reagent | HsBeclin 1 AFM1+HLM (FFW/DDD at 359–361 and L293A V296A), untagged + ZZ-HsATG14L | This work | pYO1134 | See *Supplementary file 6* for more details |
| Recombinant DNA reagent | HsBeclin1 HLM+AFM2 (L293A V296A+ Q418D F419D) | This work | pYO1190 | See *Supplementary file 6* for more details |
| Recombinant DNA reagent | His6-TEV-HsVPS34 | This work | pSM41 | See *Supplementary file 6* for more details |

## LUV generation

The LUV mixtures were mixed according to *Supplementary file 1* in a glass vial. The mixture was dried under a nitrogen gas, while rotating the tube so that a lipid film was formed on the vessel wall. The remaining solvent was evaporated by placing the glass vial in a desiccator for 60 min. The lipids were dissolved in lipid buffer (50 mM HEPES pH 8.0, 150 mM NaCl) and vortexed for 2 min. The solution was transferred to a 1.5 mL Eppendorf Tube and sonicated for 2 min in a bath sonicator, followed by 10 cycles of freeze/thaw in liquid nitrogen and a 43°C water bath. The lipid mixture was extruded 20 times through a 100 nm filter (Whatman Anotop 10 syringe filter 0.1 µm diameter, 10 mm Cat No 6809–1112). The lipid solution was then either used fresh or snap frozen in liquid nitrogen.

## GUV generation

An aliquot of 15 µL of a 1 mg/mL GUV lipid mixture in solvent was pipetted onto the Indium-Tin-Oxide (ITO)-coated side of an ITO slide (Nanion) then dried in a desiccator for 1 hr. GUVs were made in the presence of 220 µL of swelling solution (0.5 M glucose or sucrose), using a GUV maker (Vesicle Pro, Nanion) programmed for 10 Hz; 60°C; 1 Amp; 3 min rise; 68 min fall. After GUVs were generated, they were immediately removed from the ITO slide and transferred to a 1.5 mL tube, which had been incubated with 5 mg/mL BSA (Sigma A7030) for 1 hr then rinsed once with swelling solution.

## GUV immobilization

Wells of an eight well glass bottom chamber (Ibidi 80827) were treated with 100 µL of avidin solution (0.1 mg/mL avidin egg white, Life Technologies A2667 dissolved in PBS, and 1 mg/mL BSA) for 15 min then washed two times with observation buffer (25 mM HEPES pH 8.0 and 271.4 mM NaCl). An aliquot of 64 µL of observation buffer was added to the wells, followed by 48 µL GUVs then 20 µL of 0.1 mg/mL BSA-biotin (Scientific Laboratory Supplies A8549, dissolved in observation buffer). GUV immobilization was checked on the microscope, then 20 µL of 10x buffer (250 mM HEPES pH 8.0, 10 mM EGTA, 20 mM MnCl$_2$, 10 mM TCEP, and 1 mM ATP pH8.0) was added.

## Microscopy

GUVs were observed with a 63x oil immersion objective (Plan-Apochromat 63x/1.40 Oil DIC, Zeiss) on an inverted confocal microscope (Zeiss 780), using ZEN software (Zeiss). The observation chamber was immobilized on a microscope stage holder using an adhesive (Blu-Tack, Bostik). In the ZEN software, Time Series and Positions were selected. The Lissamine-rhodamine channel for GUVs was exited with a Diode-pumped solid-state (DPSS) 561 nm laser and collected with a 566–629 nm band. The Alexa Fluor 647 channel for the PX domain was exited using a HeNe 633 nm laser and collected with a 638–756 nm band. Five to six areas per well were randomly selected so that at least 15 GUVs could be analyzed. After areas were selected, an aliquot of 48 µL of a solution containing the kinase complex and the PX domain in protein dilution buffer (25 mM HEPES pH8.0, 150 mM NaCl, 1 mM TCEP, and 0.5 mg/mL BSA) was added to each well. The final concentrations of complex I, complex II or VPS34 alone are described in figures and their legends. The PX domain was used at

the final concentration of 7 µM. Images were obtained every 2–3 min for 60–120 min. For the Z-stack analysis, the Z-Stack option was ticked in the Zen software. For the Z-Stack, slices were kept, and the first and last positions were manually set and 10 slices were acquired.

## Image analysis

Images were opened with Fiji software using Bio-Formats importer plugin. For the time course analysis, partial or whole circles from GUV sections were selected using the ROI (region of interest) selection tool, and added to ROI manager. Fluorescence intensities were analyzed using a macro (GUV_intensity.ijm. See 'Confocal microscopy macro' section below). A baseline signal of the AF647 channel originating from unbound PX domain and small membrane particles or vesicles, was subtracted from all frames. To estimate this baseline, 10 regions without GUVs were selected and their intensity profile over time was obtained with the ImageJ command 'Plot Z-axis Profile'. These 10 profiles were averaged together to obtain a single temporal baseline that was later subtracted from intensity measurement at the corresponding time points. Because this baseline signal appears earlier than the actual enzyme activity, negative values are typically generated in the lag phase just before the initial rate is achieved. In the case of the kinases with very weak or no activities (*e.g.*, for the BATS domain mutant and for VPS34 alone), the signal from the baseline is stronger than the signal accumulated on the GUVs, thereby resulting in negative values after baseline subtraction. The initial rate was calculated by carrying out linear regression with Prism7 (GraphPad Software) using the points in the region of the AF647-PX signal reaction progress curve that showed a linear increase. This region was located manually and typically started after a lag phase and before the plateau of the AF647-PX signal. For the radius *versus* fluorescence intensity analysis (*Figure 2*), whole GUV circular sections were selected using Oval selection tool and added to ROI manager, and analyzed using a macro (GUV_intensity1dimension.ijm). Ten random areas for baseline subtraction were selected using the ROI selection tool, and the mean values were subtracted from the actual fluorescence intensities. Results were further analyzed using Microsoft Excel (Microsoft) and GraphPad Prism7 (GraphPad Software).

## Cell culture and transfection

Expi293 suspension cells (ThermoFisher A14527) were grown at 37°C, 8% $CO_2$, and 125 rpm shaking in Expi293 Expression Medium (ThermoFisher A1435102). Cells at a density of around $2.0 \times 10^6$ $mL^{-1}$ were transfected with plasmids at 1.1 mg/L culture, using 3 mg/L of polyethylenimine (PEI) 'MAX' (Polysciences 24765, 1 mg/mL in PBS). Cells were grown at 37°C, 8% $CO_2$, and 125 rpm shaking for 48 hr, then harvested at 3000 *g* for 20 min, flash frozen in liquid nitrogen, and stored at −80° C, until they were used.

## Flotation assay

An aliquot of 20 µL of sample was prepared containing: 1.8 mM or 1.5 mg/mL LUVs, 2 µM VPS34 complex in buffer containing 25 mM HEPES pH 8.0, 150 mM NaCl, 1 mM TCEP. The LUVs and proteins were incubated on ice for 30 min. In the meantime, a sucrose gradient was prepared. Several sucrose solutions were layered from bottom to top in a Beckman centrifuge tube (343775 Thickwall Polycarbonate Tube): 40 µL 30% sucrose solution, 52 µL 25% sucrose solution, 52 µL 20% sucrose solution. Then 16 uL of the LUV/protein sample was carefully pipetted on top of the gradient without disturbing the layers. 2.5 µL of the remaining LUV/protein was kept as an input sample. The gradient was then centrifuged for 3 hr in a TLS-55 rotor (Beckman Coulter) at 55,000 rpm and 4°C. Afterwards, 6 fractions of 26 µL were carefully collected from the top of the gradient. The input and gradient fractions were then loaded on an SDS-PAGE gel.

## Protein purification

A protein purification method for human complex I was described previously (*Ohashi et al., 2016*). Human complex II was purified in the same way as the complex I, except that the NaCl concentration is at 300 mM throughout the procedures. All complex I and complex II mutants were purified in the same conditions as the wild-type complexes. For the purification of human VPS34 alone, a plasmid (pSM41) was transformed into *E. coli* C41(DE3) RIPL cells. The transformed bacteria were cultured in 2xTY medium containing 0.1 mg/mL ampicillin at 37°C to an OD 600 of 0.8–1.0 and induced with 0.3

mM IPTG at 12°C for 15.5 hr. For a 4 L scale purification, cell pellets were resuspended in 100 mL of sonication buffer (10 mM Tris-HCl pH 8.0, 100 mM NaCl, 10 mM imidazole, 1 Complete EDTA-free Protease Inhibitor Cocktail Tablet (Roche, 11873580001), 0.1 mg/mL DNaseI, and 50 mL BugBuster (Novogen 70584) and sonicated on ice for 3 min, which was repeated three times in total. Lysates were subjected to ultracentrifugation at 100,000 $g$ for 40 min at 4°C. The supernatant was filtered through a 0.45 µm filter (Millipore, SE2M230I04), before being passed through two connected 5 mL Ni-NTA FF columns (GE Healthcare 17-5255-01). The bound protein was washed with 100 mL Ni A1 buffer (20 mM Tris pH 8.0, 300 mM NaCl, 10 mM imidazole, 2 mM β-mercaptoethanol), 100 mL Ni A2 buffer (20 mM Tris pH 8.0, 100 mM NaCl, 10 mM imidazole, 2 mM β-mercaptoethanol), and eluted with an imidazole gradient with about 80 mL of Ni B1 buffer (20 mM Tris pH 8.0, 100 mM NaCl, 300 mM imidazole, 2 mM β-mercaptoethanol). The N-terminal His6 tag was cleaved with TEV protease (made in house) and incubated overnight with gentle rocking at 4°C. On the next day, the protein was further purified on a 5 mL Heparin HP column (GE Healthcare 17040701) by first washing with 20 mL HA buffer (20 mM Tris pH 8.0, 100 mM NaCl, 2 mM DTT) and then eluting with 100 mL HB buffer (20 mM Tris pH 8.0, 2 mM DTT, 1 M NaCl). The heparin peak fractions were pooled and brought to 100 mM NaNO₃ to prevent precipitation during concentration in a 30,000 MWCO Amicon Ultra15 concentrator (Millipore UFC903024). The final purification step was carried out on a Superdex 200 16/60 gel filtration column (GE Healthcare, 17-1069-01) that was pre-equilibrated with running buffer (20 mM Tris HCl pH 8.0, 100 mM NaCl, 2 mM DTT). The peak fractions were pooled and concentrated to 5.2 mg/mL (51 µM). For the purification of human phox 40 (construct GST-TEV cleavage site-Cys-PX), a plasmid (pYO1125) was transformed into *E. coli* C41(DE3) RIPL cells. The transformed bacteria were cultured in 2xTY medium containing 0.1 mg/mL ampicillin at 37°C to an OD 600 of 0.6 and induced with 0.3 mM IPTG at 30°C for 16 hr. 6 L of cells were pelleted by 20 min centrifugation at 4,000 $g$ and the pellets were resuspended in 150 mL lysis buffer (20 mM HEPES pH 8.0, 200 mM NaCl, 1 mM TCEP, 0.05 µL/mL universal nuclease (ThermoFisher, 88702), 0.5 mg/mL lysozyme (MP Biomedicals, 195303)). The cells were sonicated 6 min on ice (10 s on/10 s off, 60% amplitude), and the lysates were spun at 30,000 $g$ for 45 min at 4°C. The supernatant was filtered through a 0.45 µm filter (Millipore, SE2M230I04), and an aliquot of 2.5 mL of washed Glutathione Sepharose beads was added (Glutathione Sepharose 4B, GE Healthcare 17-0756-05). The lysate was incubated on the beads for 45 min with gentle rolling at 6 rpm at 4°C. The beads were then transferred to a gravity flow column (Bio-Rad, 731–1550) and washed with 100 mL lysis buffer, 100 mL wash buffer (20 mM HEPES pH 8.0, 300 mM NaCl, 1 mM TCEP) and 100 mL TEV buffer (20 mM HEPES pH 8.0, 200 mM NaCl, 1 mM TCEP). The N-terminal GST tag was cleaved with TEV protease and incubated overnight with gentle rocking at 4°C. Next day, elution fractions were collected and concentrated in a 10,000 MWCO Amicon Ultra15 concentrator (Millipore, UFC901024). The concentrated protein was loaded onto a Superdex 75 16/60 gel filtration column (GE Healthcare 17-1068-01) and gel filtration was carried out in a buffer containing 20 mM HEPES pH 8.0, 200 mM KCl, 1 mM TCEP. The peak fractions were pooled and concentrated to 23.5 mg/mL (1.35 mM).

## Labelling of the PX domain

For the labeling of the PX, 1 mg of AF647 C2 Maleimide kit (Life Technologies, A20347) was dissolved in 100 µL DMSO (ThermoFischer, Catalog No. BP231-100) to final 7.7 mM. For the final labeling reaction 250 µM PX, 2.5 mM TCEP, 385 µM AF647 dye was mixed with labeling buffer (50 mM HEPES pH 7.0, 200 mM KCl) to a total volume of 1 mL, and the reaction was left for 2 hr at room temperature on a roller at 10 rpm. After the incubation, 1 mM of DTT was added and the protein was loaded on a 5 mL Heparin HP column (GE Healthcare 17040701). The column was first washed with HA buffer (50 mM HEPES PH 8.0, 100 mM KCl, 1 mM TCEP) and then eluted with HB buffer (50 mM HEPES PH 8.0, 1 M KCl, 1 mM TCEP). The peak fractions were pooled and concentrated to 6.7 mg/mL (384 µM) with ~ 20% labelling efficiency.

## Hydrogen/deuterium exchange mass spectrometry (HDX-MS)

A protein alone or a protein/lipid stock solution consisting of either 7.5 µM complex I or complex II with or without 2.2 mg/mL LUVs [18% Liver PI, 10% DOPS, 17% DOPE and 55% DOPC in LUV Buffer (25 mM HEPES pH 7.0, 100 mM NaCl and1 mM EGTA)] was incubated for 10 min at room temperature. An aliquot of 5 µL of this stock solution was mixed with 45 µL of D2O Buffer [consisting of 20

mM HEPES pH 8.0, 300 mM NaCl, 0.5 mM TCEP, 1 mg/mL LUVs (added from a 5 mg/mL LUV stock in LUV Buffer) and 74.6% D2O (Acros Organics)] for a defined period of time at 23℃. In the lipid-free experiments, the solution contained LUV Buffer instead of LUVs. The exchange reaction was quenched using 20 µL of ice cold 5 M guanidinium chloride and 8.4% Formic Acid, pH 1. The final pH of the sample was 2.2. Five time-points were produced (0.3/3/30/300/3000 s), with each exchange reaction executed in triplicate. The 0.3 s time point was obtained by carrying out an exchange for 3 s at 4℃, instead of 23℃. Each sample was immediately flash-frozen in liquid nitrogen and subsequently stored at −80℃ until analysis.

Samples were thawed and manually injected on an M-Class Acquity UPLC with HDX Manager technology (Waters) set to maintain a constant temperature of 0.1 ℃. Protein samples were digested using an in-line Enzymate Immobilised Pepsin Column (Waters) at 15 ℃ for two minutes, and were collected on a van-guard pre-column trap (Waters). Peptides were eluted from the trap onto an Acquity 1.7 µm particle, 100 mm × 1 mm C18 UPLC column (Waters), equilibrated in Pepsin-A buffer (0.1% formic acid), using a 5–36% gradient of Pepsin-B buffer (0.1% formic acid, 99.9% acetonitrile) over 20 min. Peptide Data were collected using a Waters Synapt G2 Si (Waters) over a 50 to 2000 $m/z$ range using the High-Definition MS$^e$ data acquisition mode fitted with an ESI source.

Peptide identification was conducted using the ProteinLynx Global Server (PLGS, Waters, U.K.). Peptides were identified from three non-deuterated samples for complex I and for complex II. Deuterated peptides were analysed using DynamX 3.0 software (Waters, U.K.). Peptide inclusion criteria was a minimum intensity of 5000, a minimum sequence length of 5 amino acids, a minimum of 0.1 products per amino acid, a maximum MH+ error of 5 ppm, and a positive identification meeting these criteria in at least 2 of the three non-deuterated files. An initial automated spectral processing step was conducted by DynamX followed by a manual inspection of individual peptides for sufficient quality. A table of all peptides included in the analysis and their quality assessment statistics are available in the *Supplementary files 4* and *5*. The HDX analysis in this manuscript complies with the community agreed guidelines (*Masson et al., 2019*).

## SEC-MALS

The molecular masses of human complexes I and II were determined in solution using size exclusion chromatography coupled to multi-angle light scattering (SEC-MALS). Measurements were performed using a Wyatt Heleos II 18 angle light scattering instrument coupled to a Wyatt Optilab rEX online refractive index detector. Samples for analysis (human complex I at 1 mg/mL) and complex II (0.8 mg/mL) were resolved on a Superose 6 10/300 analytical gel filtration column (GE Healthcare) coupled to an Agilent 1200 series LC system running at 0.5 mL/min before then passing through the light scattering and refractive index detectors in a standard SEC-MALS format. Protein concentration was determined from the excess differential refractive index based on 0.186 ΔRI for 1 g/mL or with the sequence based UV extinction coefficient determined in ProtParam. The measured protein concentration and scattering intensity were used to calculate molecular mass from the intercept of a Debye plot using Zimm's model as implemented in the Wyatt ASTRA software.

The experimental setup was verified using a BSA standard run with the same sample volume. The monomer peak was used to check mass determination and to evaluate interdetector delay volumes and band broadening parameters that were subsequently applied during analysis of complexes I and II.

## Confocal microscopy macro

The mean intensity for each label on the GUVs' membranes was extracted using a custom ImageJ macro. The rings corresponding to the 2D section of the GUVs were segmented by applying a threshold to the response of a difference of a Gaussian filter applied for each channel and at each time point. The logical union of the masks of the two segmented channels was then computed in order to obtain a unique mask for each time point. After smoothing, the inner part of the rings was filled and an internal morphological gradient provided the contours of the GUVs with a calibrated thickness. Finally, the membrane of the GUVs are obtained as the intersection of these thick contours and the original segmentation, allowing to discard any potential spurious signal within the GUVs. Within a set of manually selected regions of interest, the mean intensities of the signal in the

segmented membranes for each channels and time points were recorded and assigned to the corresponding region thus circumventing the problem of tracking possibly complex regions.

## Lipid mixtures

All GUV and LUV lipid mixtures are described in *Supplementary files 1* and *2*.

## Plasmids

All plasmids are described in *Supplementary file 6*.

## Acknowledgements

We thank Aurélien Roux and Annika Hohendahl (University of Geneva) for help with assay development. We thank Olga Perisic for help with protein expression and purification, advice on assay development and discussions, Maria Daly for help with exploring flow cytometry-based assays, Lufei Zhang for purifying VPS34 alone, Sarah Maslen and Mark Skehel for assistance with HDX-MS and Conrad Weichbrodt of Nanion Technologies for advice on preparation of GUVs. The work was supported by the MRC (File reference number MC_U105184308 to RLW) and Cancer Research UK (Programme grant C14801/A21211 to RLW). ST was supported by an MRC studentship. GRM was supported by AstraZeneca/LMB Blue Sky Initiative [MC-A024-5PF9L] and by a Henslow Research Fellowship from The Cambridge Philosophical Society and St Catharine's College, Cambridge.

## Additional information

### Funding

| Funder | Grant reference number | Author |
| --- | --- | --- |
| Medical Research Council | MC_U105184308 | Roger L Williams |
| Cancer Research UK | C14801/A21211 | Roger L Williams |
| Medical Research Council | MC-A024-5PF9L | Roger L Williams |

The funders had no role in study design, data collection and interpretation, or the decision to submit the work for publication.

### Author contributions

Yohei Ohashi, Conceptualization, Data curation, Formal analysis, Supervision, Validation, Investigation, Visualization, Methodology, Writing - original draft, Writing - review and editing; Shirley Tremel, Roger L Williams, Conceptualization, Resources, Data curation, Software, Formal analysis, Supervision, Funding acquisition, Validation, Investigation, Visualization, Methodology, Writing - original draft, Project administration, Writing - review and editing; Glenn Robert Masson, Data curation, Formal analysis, Supervision, Validation, Investigation, Methodology, Writing - review and editing; Lauren McGinney, Christopher M Johnson, Izabella Niewczas, Data curation, Formal analysis, Investigation, Methodology, Writing - review and editing; Jerome Boulanger, Software, Methodology, Writing - review and editing; Ksenia Rostislavleva, Investigation, Methodology, Writing - review and editing; Jonathan Clark, Data curation, Formal analysis, Supervision, Investigation, Methodology, Writing - review and editing

### Author ORCIDs

Yohei Ohashi https://orcid.org/0000-0002-2288-130X
Shirley Tremel https://orcid.org/0000-0002-4077-0021
Glenn Robert Masson http://orcid.org/0000-0002-1386-4719
Roger L Williams https://orcid.org/0000-0001-7754-4207

### Decision letter and Author response

Decision letter https://doi.org/10.7554/eLife.58281.sa1
Author response https://doi.org/10.7554/eLife.58281.sa2

## Additional files

### Supplementary files

- Supplementary file 1. Composition of GUVs and LUVs used for VPS34 assays.
- Supplementary file 2. Sources of lipids for GUVs and LUVs.
- Supplementary file 3. Reaction time courses with error bars.
- Supplementary file 4. Summary of HDX results for complex I with Lipids.
- Supplementary file 5. Summary of HDX results for complex II with Lipids.
- Supplementary file 6. Summary of plasmids used to express the proteins and protein complexes.
- Transparent reporting form

### Data availability

All data generated or analysed during this study are included in the manuscript and supporting files.

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
