## [Decision Letter]

**Acceptance summary:**

In this paper the authors studied the *in vitro* activity of the PI3K kinase complexes I and II -which share the VPS34 catalytic core but differ in the regulatory units. They reported large changes in the activity when changing membrane properties (such as lipid unsaturation, curvature or overall electrostatics). Technically, this is a superb study, which provides a clear-cut demonstration of the importance of membrane physical parameters in the activation of these kinases.

**Decision letter after peer review:**

Thank you for submitting your article "Membrane characteristics tune activities of endosomal and autophagic human VPS34 complexes" for consideration by *eLife*. Your article has been reviewed by three peer reviewers, including Felix Campelo as the Reviewing Editor and Reviewer #1, and the evaluation has been overseen by Vivek Malhotra as the Senior Editor.

The reviewers have discussed the reviews with one another and the Reviewing Editor has drafted this decision to help you prepare a revised submission.

The editors have judged that your manuscript is of interest, but as described below additional experiments are required before we proceed further for publication. We would like to draw your attention to changes in our revision policy that we have made in response to COVID-19 (https://elifesciences.org/articles/57162). First, because many researchers have temporarily lost access to the labs, we will give authors as much time as they need to submit revised manuscripts. We are also offering, if you choose, to post the manuscript to bioRxiv (if it is not already there) along with this decision letter and a formal designation that the manuscript is "in revision at *eLife*". Please let us know if you would like to pursue this option. (If your work is more suitable for medRxiv, you will need to post the preprint yourself, as the mechanisms for us to do so are still in development.)

Summary:

In this study, the group of Roger L. Williams performed an extensive biochemical characterization of two PI3K kinase complexes, I and II, which both include the same VPS-34 catalytic core but have different regulatory units that confer them different avidity for membrane-bound organelles. Complex I, which contains ATG14L, is involved in autophagy whereas Complex II, which contains the subunit UVRAG, is involved in endocytosis. The experimental strategy used here is novel, and allows the authors to address membrane physical parameters important for complex I and/or complex II activity and membrane recruitment in a quite extensive and exquisite manner. They use giant unilameller vesicles incubated with a fluorescent PX domain that is specifically recruited by PI3P, the product of the PI3 kinase activity of the two complexes. Because liposome binding of this fluorescent reporter is weakly affected by bulk membrane parameters, the PX domain is a fair reporter of PI3P production and enables real time measurements at the GUV surface. The authors reported very large changes in complex I and/or complex II activity by changing lipid membrane unsaturation, curvature or overall electrostatics. However, the two complexes do not respond in the same manner to these parameters, suggesting interesting hypothesis for their differential localization and activation in cells. Technically, this is a superb study. The effects are large and confirmed by different independent analysis procedures (e.g. GUV imaging vs LUV flotation). The rationale is thorough and easy to follow. The Hydrogen/deuterium exchange mass spectrometry experiments are very informative. Conceptually, some findings were expected, but the work provides a clear-cut demonstration of the importance of membrane physical parameters in the activation of these kinases.

Essential revisions:

1) A general concern is in the physiological significance of the results. Is it possible to use lipids compositions similar to the ER for complex I and endosomes for complex II to validate some of the key conclusions (especially the effect of BATS domain and BARA domain)? Alternatively, is it possible to perform some experiments in the presence of Rab5-GTP attached to GUVs by a hisx6 tag? (using Ni-NTA lipids). It would be interesting to see if complex II in the presence of Rab5 can reach the same activity as complex I in the absence of Rab5-GTP but with an optimal lipid composition (e.g. unsaturation, curvature).

2) Another concern (which is probably a semantic) is how and whether the authors can differentiate between "activity" (in the sense of actual enzymatic activity of the protein) and "binding affinity" (that is, the amount of proteins recruited per unit area to a GUV. Particularly, the authors write "This indicates that complex I is more active than complex II because complex I binds membranes more tightly than complex II.". Is the measured increase in PI3P amounts solely based on increased fraction of the kinase present in those GUVs? Or the lipid microenvironment where the kinase sits also influences its intrinsic enzymatic activity? Protein binding experiments are shown only in Figure 1D. It would have been interesting to have such parallel experiments for the other lipid compositions (especially in Figure 3), but this is not something we expect in the revision. In any case, the authors should explain well what their conclusions are actually showing (binding/activity or a combination of both).

3) The last set of experiments using PI(4)P or PI(4,5)P_2_ GUVs show very modest effects of these lipids. However, the authors interpret these experiments in a rather exaggerated manner. It seems that these experiments simply show that complex I and II are not sensitive to these phosphoinositides.

---

## [Author Response]

Essential revisions:1) A general concern is in the physiological significance of the results. Is it possible to use lipids compositions similar to the ER for complex I and endosomes for complex II to validate some of the key conclusions (especially the effect of BATS domain and BARA domain)? Alternatively, is it possible to perform some experiments in the presence of Rab5-GTP attached to GUVs by a hisx6 tag? (using Ni-NTA lipids). It would be interesting to see if complex II in the presence of Rab5 can reach the same activity as complex I in the absence of Rab5-GTP but with an optimal lipid composition (e.g. unsaturation, curvature).

The reviewer’s suggestion is useful. Unfortunately, determining an accurate lipid composition of a cellular compartment is a greatly needed, evolving area of study that is progressing in development of better methods of isolating compartments and improved MS techniques for analysing lipid populations. As described at the beginning of the section “Human complex I is more active than complex II” in the Results, our base lipid composition is similar to ER membranes. We chose this because ER membranes are interconnected to all organelles including endosomes, and lipids are shuttled between the ER and other compartments (Prinz et al., 2020). It is believed that early endosomes share a similar lipid composition with the plasma membrane, since they are both part of the same recycling pathway (Bissig and Gruenberg, 2013). Like the plasma membrane, early endosomes are rich in cholesterol and sphingomyelin (Arumugam and Kaur, 2017; Kobayashi et al., 2002). Therefore, in the revised manuscript we present the activities of complexes I and II when the initial composition is enriched with 10% cholesterol and 10% sphingomyelin (Figure 1—figure supplement 2 in the revised manuscript). This revealed that even with this composition, complex I remains more active than complex II. While this early endosome mimic did not make complex II have equal activity to complex I, we know that the lipid composition can affect the relative activities of the two complexes, as we showed in Figure 4G for 25% PS-containing GUVs where the activities are equal. Although this is an unrealistic PS content for early endosomes, it is likely that the higher PS content of early endosomes along with Rab5 will favour complex II activity. We have described the results of our cholesterol/sphingomyelin assays in the same section of the revised manuscript.

The BATS experiments in Figure 7, and the BARA experiments for complexes I and II in Figure 8 were performed on the ER-mimicking DO base lipid GUVs, where it is physiologically relevant for the BATS domain of complex I. Regarding the complex II experiments for the BARA characterisation, the effects of the aromatic finger 1 mutation on activity and membrane binding have been reported using several different lipid compositions and sizes (Chang et al., 2019; Huang et al., 2012; Rostislavleva et al., 2015), indicating that the importance of the mutations are relevant for a range of lipid compositions.

Although determining the influence of Rab5 is very important, and indeed it is one of our future objectives, this will measure the effect of Rab5 on the complexes, rather than the effect of lipid properties on them. Therefore we feel this is beyond the scope of this manuscript.

2) Another concern (which is probably a semantic) is how and whether the authors can differentiate between "activity" (in the sense of actual enzymatic activity of the protein) and "binding affinity" (that is, the amount of proteins recruited per unit area to a GUV. Particularly, the authors write "This indicates that complex I is more active than complex II because complex I binds membranes more tightly than complex II.". Is the measured increase in PI3P amounts solely based on increased fraction of the kinase present in those GUVs? Or the lipid microenvironment where the kinase sits also influences its intrinsic enzymatic activity? Protein binding experiments are shown only in Figure 1D. It would have been interesting to have such parallel experiments for the other lipid compositions (especially in Figure 3), but this is not something we expect in the revision. In any case, the authors should explain well what their conclusions are actually showing (binding/activity or a combination of both).

We appreciate the reviewer’s point. We have clarified our writing to distinguish that what we are generally measuring is activity. Increased membrane binding is one possible mechanism of increasing activity. We have restated our interpretation of Figure 1 in the section of the results “Human complex I is more active than complex II” and in the Results section “The ATG14L BATS domain is critical for the activity of complex I on membranes”, describing Figure 7. For the other figures we believe that we had stated what the results are showing.

3) The last set of experiments using PI(4)P or PI(4,5)P_2_ GUVs show very modest effects of these lipids. However, the authors interpret these experiments in a rather exaggerated manner. It seems that these experiments simply show that complex I and II are not sensitive to these phosphoinositides.

The reviewer is correct. The effects of the polyphosphoinositides are less substantial, however, they may be more dynamic. The polyphosphoinositides are, in general, minor components of lipids, but the rapidly respond to signalling. We have revised and simplified this in the Discussion section in the paragraph before the last paragraph. We have also modified the Results section “Phosphoinositides have modest effects on activities of complexes I and II”.